# THE WORLD IS CHANGING: IMPROVING FAIR TRAINING UNDER CORRELATION SHIFTS

## ABSTRACT

Model fairness is an essential element for Trustworthy AI. While many techniques for model fairness have been proposed, most of them assume that the training and deployment data distributions are identical, which is often not true in practice. In particular, when the bias between labels and sensitive groups changes, the group fairness of the trained model is directly influenced and can worsen. We make two contributions for solving this problem. First, we analytically show that existing in-processing fair algorithms have fundamental limits in accuracy and group fairness. We introduce the notion of *correlation shifts*, which can explicitly capture the change of the above bias. Second, we propose a novel pre-processing step that samples the input data to reduce correlation shifts and thus enables the in-processing approaches to overcome their limitations. We formulate an optimization problem for adjusting the data ratio among labels and sensitive groups to reflect the shifted correlation. A key advantage of our approach lies in *decoupling* the roles of pre-processing and in-processing approaches: correlation adjustment via pre-processing and unfairness mitigation on the processed data via in-processing. Experiments show that our framework effectively improves existing in-processing fair algorithms w.r.t. accuracy and fairness, both on synthetic and real datasets.

## 1 INTRODUCTION

Model fairness is becoming indispensable in many artificial intelligence (AI) applications to prevent discrimination against specific groups such as gender, race, or age (Feldman et al., 2015; Hardt et al., 2016) or individuals(Dwork et al., 2012a). In this work, we focus on group fairness, and there are three prominent group fairness approaches: pre-processing, where training data is debiased; in-processing, where model training is tailored for fairness; and post-processing, where the trained model's output is modified to satisfy fairness – see more related works discussed in Sec. 6.

While fairness in-processing approaches are commonly used to mitigate unfairness, most of them make the limiting assumption that the training and deployment data distributions are the same (Zafar et al., 2017a; Zhang et al., 2018; Roh et al., 2021). However, the two distributions are usually different, especially in terms of data biases (Wick et al., 2019; Maity et al., 2021). For example, a recent work shows that the bias amounts likely differ between previously collected data and recently collected data (Ding et al., 2021). Moreover, when the data bias changes, the fairness and accuracy of the trained model are now unpredictable at deployment, as the above assumption is broken.

In this work, we introduce the notion of *correlation shifts* between the label y and group attribute z in the data to systematically address the data bias changes. Although several works have been recently proposed to investigate fair training on different types of distribution shifts, including covariate and concept shifts (Singh et al., 2021; Mishler & Dalmasso, 2022), they usually do not explicitly consider bias changes between y and z. In comparison, our correlation shifts enables us to theoretically analyze how exactly data bias changes affect fair training – see how correlation shift compares with other types of distribution shifts in Sec. 6.

For fair training under correlation shifts, we first 1) analyze the fundamental accuracy and fairness limits of in-processing approaches with the fixed distribution assumption using the notion of correlation in the data and then 2) design a novel pre-processing step to boost the performances of in-processing approaches under the correlation shifts. We show that existing in-processing fair algorithms are indeed limited by the training distribution and may perform poorly on the deployment

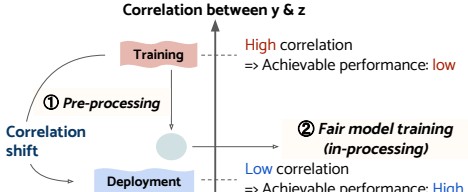

(a) A high-level workflow under correlation shifts.

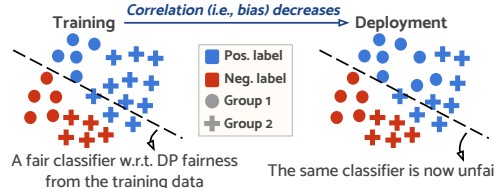

(b) A toy example for illustrating the impact of correlation shifts (i.e., bias changes) on the trained classifier.

Figure 1: The central axis in the left figure represents the correlation between the label y and sensitive group attribute z. The correlation of training data is usually higher than that of deployment data. In Sec. 3, we show that the correlation determines the achievable performance of fair training. Thus, we first run our pre-processing and then apply existing fair algorithms on the processed data to address the correlation shift and improve the performances of fair training. Not addressing the correlation shift may result in reduced performance as shown in the right figure – see details in Sec. 2.

distribution. In particular, a high (y, z)-correlation results in a poor accuracy-fairness tradeoff for any fair training. Therefore, as most in-processing fair algorithms assume identical training and deployment distributions, there is no guarantee their performances on the training data carry over to the deployment data. Based on the theoretical analysis, we propose a pre-processing step for reducing the shifted correlation by taking samples of (y, z)-classes. Using a possible range of the shifted correlations, we solve an optimization problem that finds the new data ratio among (y, z)-classes to adjust the correlation for the shift, which gives in-processing approaches a better opportunity to perform well. The new data is then used as the input of any fair algorithm.

A key advantage of our framework is *the decoupling of pre-processing and in-processing for unfairness mitigation* where the pre-processing adjusts the correlation while the in-processing performs the rest of the unfairness mitigation, as described in Figure 1a. We note that our pre-processing aims to boost the performances of in-processing approaches based on our theoretical analysis, whereas existing pre-processing approaches for fairness simply remove the bias in the data and are not designed to explicitly benefit the in-processing approaches – see Sec. 5.1 for details. Our framework thus takes the best of both worlds of pre- and in-processings where (1) pre-processing solves the data problems, and (2) in-processing performs its best on the improved data. Our framework is not only useful for improving the fairness of a single metric, but can also be extended to support multiple metrics.

In our experiments, we verify our theoretical results and demonstrate how our framework outperforms state-of-the-art pre-processing and in-processing baselines. Experiments on both synthetic and real-world datasets (COMPAS (Angwin et al., 2016) and AdultCensus (Kohavi, 1996)) show that our framework effectively improves the accuracy and fairness performances of the state-of-the-art in-processing approaches (Zafar et al., 2017a; Zhang et al., 2018; Roh et al., 2021) under correlation shifts. Also, our framework performs better than two-step baselines that first run an existing pre-processing approach (Kamiran & Calders, 2011) and then an in-processing approach. We also show that our framework is still beneficial when we do not know the exact range of the shifted correlations.

**Summary of Contributions**  **(1)** We introduce the notion of correlation shifts, which is important to connect the data bias changes and behaviors of fair training. **(2)** Using the notion of correlation, we theoretically show that existing in-processing fair algorithms are limited by the training distribution and may perform poorly on the deployment distribution. **(3)** We propose a novel pre-processing step to boost the performances of fair in-processing approaches. **(4)** We demonstrate that our framework effectively improves the performances of the state-of-the-art fair algorithms under correlation shifts.

## 2 LIMITATIONS OF FAIR TRAINING WITH FIXED DISTRIBUTION ASSUMPTION

Most in-processing approaches for fairness assume that the training and deployment distributions are the same, which means that they assume the same level of bias as well. However, data bias may shift over time as confirmed by recent studies (Wick et al., 2019; Maity et al., 2021; Ding et al., 2021), which means that the deployment data may actually have a different bias than the training data.

In fair training, the data bias reflects the relationship between a label y and sensitive group attribute z. For example, if all positive labels are in the same group, the data can be considered highly biased. Conversely, if the labels are randomly assigned to all groups, the data can be considered unbiased.

A bias change in the deployment data may have an adverse affect on a trained model's performance. Figure 1b shows a toy example that illustrates how a fair classifier's performance is affected by a bias change during deployment. Here, the bias can be expressed via correlation, and we discuss their

relationship in Sec. 3. The training distribution is biased, where Group 2 has more positive labels than Group 1. On the other hand, in the deployment distribution, the bias decreases as the positive labels are equally distributed for each group. Suppose we train a fair classifier on the training data as shown on the left side where the DP fairness is perfect (i.e., $\Pr(\hat{y}=1)$ are the same for the groups), but the accuracy is not as a result. On the deployment data, the DP worsens while the accuracy still remains imperfect. The underlying problem is that the classifier was trained with a different bias in mind.

In the next section, we formalize the notion of bias change and provide a theoretical analysis and simulation results that explain the above observations with the following notation and fairness metrics.

**Notation & Fairness Metrics**   Let $\theta$ be the model weights, $x \in \mathbb{X}$ be the input feature to the model, $y \in \mathbb{Y}$ be the true label, and $\hat{y} \in \mathbb{Y}$ be the predicted label where $\hat{y}$ is a function of $(x, \theta)$. Let $z \in \mathbb{Z}$ be a sensitive group attribute, e.g., gender or race. We assume a binary setting ($\mathbb{Y} = \mathbb{Z} = \{0, 1\}$). We focus on the prominent group fairness metrics, demographic parity (DP) (Feldman et al., 2015) and equalized odds (EO) (Hardt et al., 2016), where DP is achieved when the positive prediction rates are the same for the groups (i.e., $\Pr(\hat{y}=1|z=1)= \Pr(\hat{y}=1|z=0)$) and EO is achieved when the label-wise accuracies are the same for the groups (i.e., $\Pr(\hat{y}=y|y=y, z=1)= \Pr(\hat{y}=y|y=y, z=0), \forall y \in \{0, 1\}$).

## 3   FAIR TRAINING UNDER CORRELATION SHIFTS

To systematically study the effects of data bias changes, we first analyze the achievable performances of fair training via a *correlation between y and z that can be used to explicitly measure data bias regarding sensitive groups* (Sec. 3.1) and discuss the limitations of fair in-processings when this correlation shifts (Sec. 3.2). We define correlation as follows:

**(y, z)-correlation**   Data bias can be represented via the correlation between the label y and the sensitive group attribute z, where the correlation represents a statistical relationship between two random variables. We thus define *(y, z)-correlation* using Pearson's correlation coefficient (Rodgers & Nicewander, 1988) $\rho_{yz} = \frac{Cov(y,z)}{\sigma(y)\sigma(z)}$, which is known to effectively capture biases in real-world scenarios. Here, $Cov(\cdot)$ is the covariance, and $\sigma(\cdot)$ is the standard deviation. As we assume y and z are binary, we can express $\rho_{yz}$ as follows (Cohen & Cohen, 1975):

$$\rho_{yz} = \frac{\Pr(y=1, z=1)\Pr(y=0, z=0) - \Pr(y=1, z=0)\Pr(y=0, z=1)}{\sqrt{\Pr(y=1)\Pr(y=0)\Pr(z=1)\Pr(z=0)}}.$$

### 3.1   ACHIEVABLE PERFORMANCE ANALYSIS VIA CORRELATION

We identify the fundamental accuracy and fairness limits of fair training based on the (y, z)-correlation of the data. We first analyze the most common case of improving the fairness w.r.t. a single fairness metric. We then extend our analysis to the more complicated case of improving the fairness w.r.t. multiple metrics, which is important for capturing various social contexts, but has been seldom studied in the literature. In both cases, we show that the (y, z)-correlation determines the achievable performance of fair training.

**CASE 1 – Fair Training w.r.t. a Single Metric**   When improving group fairness, fair training is known to face an accuracy-fairness tradeoff, where the accuracy is sacrificed to make the model fairer. Recently, Menon & Williamson (2018) investigate that the accuracy-fairness tradeoff w.r.t. demographic parity is affected by how much y and z are *aligned* in the data. For example, if the y and z values are identical for all examples, achieving high fairness may require low accuracy. In contrast, if y and z are randomly set, fairness and accuracy can be achieved together. The following proposition shows this previous work's result.

**Proposition 1** (From Menon & Williamson (2018))**.** *(Informal) When a model is trained w.r.t. demographic parity, a high alignment between* y *and* z *leads to a worse accuracy-fairness tradeoff.*

For our purposes, we infer a similar relationship using (y, z)-correlation based on Proposition 1. The following lemma makes a connection between the (y, z)-correlation and the conditional probabilities of y given z under some conditions.

**Lemma 2.** *If the marginal probabilities of* y *and* z *(i.e.,* $\Pr(y = y)$ *and* $\Pr(z = z)$*) remain the same, the (y, z)-correlation* $\rho_{yz}$ *is proportional to the difference between the conditional probabilities of* y *given different* z *values (i.e.,* $\Pr(y = 1|z = 1) - \Pr(y = 1|z = 0)$*).*

The proof for Lemma 2 can be found in Sec. A.1. By applying Lemma 2 to Proposition 1, we can infer that the $(y, z)$-correlation also determines the accuracy-fairness tradeoff under certain conditions. According to the previous work (Menon & Williamson, 2018), alignment is defined as how many examples in each group have a specific label. We can thus measure alignment as $\Pr(y = y | z = 1) + \Pr(y = y' | z = 0)$ where $y \neq y'$ and $y, y' \in \{0, 1\}$. Then, we can convert this term into the difference between the conditional probabilities of y given different z values, which is used in Lemma 2 – see details in Sec. A.2. As a result, we derive the following corollary, which shows that the $(y, z)$-correlation determines the achievable accuracy-fairness tradeoff.

**Corollary 3.** *When a model is trained w.r.t. demographic parity, and the marginal probabilities of* y *and* z *remain the same, the achievable accuracy-fairness tradeoff of the model is determined by the* ($y, z$)*-correlation. The higher the correlation, the worse the accuracy-fairness tradeoff.*

**Remark 4.** *We can relax the assumption that marginal probabilities are fixed in Corollary 3. When the marginal probabilities of* y *and* z *change up to* $\gamma_y$ *and* $\gamma_z$, *respectively, Corollary 3 can be generalized as follows: the achievable accuracy-fairness tradeoff is determined by* $\rho_{y,z} \cdot \eta$, *where* $\rho_{y,z}$ *is the* ($y, z$)*-correlation and* $\eta \in [\sqrt{\frac{\Pr(y=1) - \gamma_y - (\Pr(y=1) + \gamma_y)^2}{\Pr(z=1) + \gamma_z - (\Pr(z=1) - \gamma_z)^2}}, \sqrt{\frac{\Pr(y=1) + \gamma_y - (\Pr(y=1) - \gamma_y)^2}{\Pr(z=1) - \gamma_z - (\Pr(z=1) + \gamma_z)^2}}]$. *Thus, the higher the* $\rho_{y,z} \cdot \eta$, *the worse the accuracy-fairness tradeoff. In our framework, we actually allow the marginal probabilities of* y *and* z *to change up to* $\gamma_y$ *and* $\gamma_z$ *– see details in Sec. 4.1.*

**Simulation** We now confirm our theoretical observations via a simulation – see details on the setting in Sec. B.1. The left plot in Figure 2 shows the accuracy-unfairness performances of classifiers on two synthetic datasets with low and high correlations when using DP for measuring fairness (the plot on the right will be explained later). We measure unfairness where a lower value indicates better fairness (i.e., perfectly fair when the value is 0) – see the exact metrics in Sec. 5. We generate various synthetic classifiers on the two datasets to show the full range of possible model performances. The blue dots (red crosses) are the classifiers on the dataset with low (high) correlation. As a result, low correlation results in better accuracy-fairness tradeoffs (i.e., close to the bottom right). We will discuss the black stars and squares in Sec. 3.2.

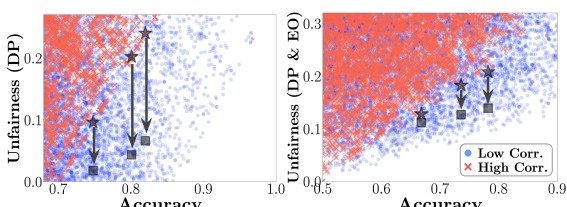

Figure 2: Simulation results of accuracy-unfairness (left: DP and right: DP & EO) performances of various classifiers on two synthetic datasets that have low and high (y, z)-correlations. Each blue dot (red cross) indicates a single classifier's performance on the low (high) correlation data. The lower the unfairness value, the better. For each dataset, we generate enough classifiers to show the full range of possible performances. As a result, low correlation enables classifiers to attain better accuracy-fairness tradeoffs (i.e., close to the bottom right), regardless of supporting single or multiple fairness metrics. Also, the optimal classifiers trained on high correlation data (black stars) have suboptimal performances on the low correlation data (black squares).

**Remark 5.** *We note that Corollary 3 does not necessarily apply for equalized odds (EO), which is achieved when the accuracies conditioned on the true label are the same for the groups. In theory, a perfect classifier can achieve perfect fairness w.r.t. EO, so the (y, z)-correlation does not determine the limits of the accuracy-fairness tradeoff w.r.t. EO. However, classifiers are not perfect in practice, and we empirically observe that higher (y, z)-correlation leads to a worse accuracy-fairness tradeoff w.r.t. EO as well – see the empirical results in Sec. B.2.*

**CASE 2 – Fair Training w.r.t. Multiple Metrics** Beyond the single metric scenario, we now extend our analysis to support multiple fairness metrics together. Although supporting multiple metrics is necessary to address fairness under various social contexts, most fairness works do not address this problem. A major challenge is that group fairness metrics are known to be mutually exclusive, which means that these metrics cannot be perfectly satisfied together, unless the data is completely unbiased (Barocas et al., 2019; Kleinberg et al., 2017). Interestingly, our (y, z)-correlation provides an opportunity to support multiple fairness metrics together as much as possible in a principled fashion. We focus on supporting two metrics and leave the support of more metrics as future work.

We first analyze when a model can improve both demographic parity (DP) and equalized odds (EO) to get hints on the relationship between the (y, z)-correlation and improving fairness w.r.t. two metrics. Here, we focus on the $\varepsilon$-fairness of both metrics, which is a relaxed version of perfect fairness (Barocas et al., 2019), where $\varepsilon$ indicates the unfairness level of the model.

For example, we can define $\varepsilon$-DP as $|\Pr(\hat{y} = 1|z = 1) - \Pr(\hat{y} = 1|z = 0)| \le \varepsilon$ and $\varepsilon$-EO as $|\Pr(\hat{y} = y|y = y, z = 1) - \Pr(\hat{y} = y|y = y, z = 0)| \le \varepsilon$. Note that $\varepsilon \in [0, 1]$ and a lower $\varepsilon$ means higher fairness. The following proposition shows when a model can achieve both $\varepsilon$-DP and $\varepsilon$-EO.

**Proposition 6** ($\varepsilon$-DP & $\varepsilon$-EO). *Let a model achieve both $\varepsilon$-DP and $\varepsilon$-EO. Then, the following inequality holds:* $|\Pr(y = y|z = z) - \Pr(y = y|z = z')| \ |\Pr(\hat{y} = y|y = y, z = z) - \Pr(\hat{y} = y|y = y', z = z)| \le 2\varepsilon, y \neq y', z \neq z', y, y', z, z' \in \{0, 1\}$.

The proof for Proposition 6 can be found in Sec. A.3. Here, the achievable fairness level $\varepsilon$ is affected by the data bias, which is represented by the difference between the conditional probabilities of y given z (i.e., $|\Pr(y = y|z = z) - \Pr(y = y|z = z')|$). For example, if the data is highly biased, a model cannot achieve high fairness (i.e., low $\varepsilon$) unless the second term on the left side (i.e., $|\Pr(\hat{y} = y|y = y', z = z) - \Pr(\hat{y} = y|y = y, z = z)|$) decreases. However, the second term is related to the accuracy, where lowering it to zero causes the model to make random predictions or predictions that are all the same. Thus, Lemma 2 and Proposition 6 give the following corollary.

**Corollary 7.** *When a model is trained w.r.t. both DP and EO, and the marginal probabilities of y and z do not change, the achievable fairness of the model is determined by the (y, z)-correlation if the accuracy does not change. The higher the correlation, the lower the achievable fairness.*

Similar to the single-metric case, we confirm the theoretical result of supporting multiple metrics via the simulation in the right plot in Figure. 2. Again, we observe that low correlation enables classifiers to attain better accuracy-fairness tradeoffs. We also leave discussions for improving fairness w.r.t. another prominent metric called predictive parity (PP) (Berk et al., 2021) together with DP and EO in Secs. A.4 ($\varepsilon$-PP and $\varepsilon$-DP) and A.5 ($\varepsilon$-EO and $\varepsilon$-PP).

## 3.2 EFFECTS OF CORRELATION SHIFTS

As the performance ranges of fair training are different according to the data correlation, a fair classifier learned on some training data does not necessarily have optimal performance on deployment data with a different correlation. We say there is a *correlation shift* between two data distributions $D_1$ and $D_2$ when $|\rho_{yz}^{D_1} - \rho_{yz}^{D_2}| \neq 0$, where $\rho_{yz}^{D_i}$ indicates the (y, z)-correlation of $D_i$. Here, the correlation *shift* reflects the bias *change* between data distributions.

We revisit our previous simulation in Figure 2 that helps to understand how the correlation shifts affect fair training. We consider a correlation shift from a training distribution with high correlation to a deployment distribution with low correlation. We start from a few optimal classifiers trained on the high-correlated data, which are denoted as black stars. These classifiers unfortunately have suboptimal performances on the low-correlation deployment data (black squares). Hence, in-processing approaches that assume the same training and deployment data distributions lose the chance of achieving better fairness and accuracy in the deployment distribution under correlation shifts.

Based on our theoretical and simulation results, we design a pre-processing approach using (y, z)-correlation as a tuning knob to give existing in-processing approaches a better opportunity to achieve maximum accuracy and fairness performances in the following section.

## 4 FRAMEWORK

To improve the performance of fair training in the presence of correlation shifts, we propose a novel fair training framework that consists of two main parts: applying a pre-processing approach for reflecting the shifted correlation and utilizing any existing in-processing algorithms for fair training on top of the improved data. We first design an optimization problem for finding a new data distribution that follows the shifted correlation by adjusting the (y, z)-class ratios. Then, we explain our overall training process and extensions to support similar distributions. We note that our framework implicitly assumes that the distribution of input feature x does not change drastically – see details in Sec. A.6.

## 4.1 OPTIMIZATION

We design an optimization that finds the best (y, z)-class ratios given a (y, z)-correlation $\rho_{yz}$ by using Lemma 2, which shows that $\rho_{yz}$ is proportional to the conditional probability difference under some assumptions. Note that the conditional probability can be written using the class weights. Let the original data ratio of each (y=$y$, z=$z$)-class be $w_{y=y, z=y}$. Let the new data ratio be $w'_{y=y, z=y}$, which

is required to satisfy the shifted correlation of the deployment data. Note that $\sum_{\forall y,z} w'_{\text{y}=y,\text{z}=z} = 1$. Also, let $c$ be the correlation constant that is the difference between the conditional probabilities of y given z in the deployment data. Let $[\alpha, \beta]$ be the range of the correlation constant $c$ (i.e., $c \in [\alpha, \beta]$). The range may be known in advance or constructed by using distribution estimation techniques (Huang et al., 2006; Zhang et al., 2013) for computing the shifted correlation and generating a range using its confidence interval. We use a relaxed version of the assumption in Lemma 2 where the marginal probabilities of y and z can change by up to $\gamma_\text{y}$ and $\gamma_\text{z}$, respectively, as discussed in Remark 4. With these conditions, we set up a problem whose goal is to minimize the squared differences between the original and new data ratios to help reduce the information loss, similar to other pre-processing approaches (Zemel et al., 2013; Quadrianto et al., 2019):

$$\min_{w'} \sum_{\forall y,z} (w_{\text{y}=y,\text{z}=z} - w'_{\text{y}=y,\text{z}=z})^2$$

$$\text{s.t.} \quad \alpha \leq \frac{w'_{\text{y}=1,\text{z}=1}}{w'_{\text{y}=1,\text{z}=1} + w'_{\text{y}=0,\text{z}=1}} - \frac{w'_{\text{y}=1,\text{z}=0}}{w'_{\text{y}=1,\text{z}=0} + w'_{\text{y}=0,\text{z}=0}} \leq \beta,$$

$$|(w'_{\text{y}=1,\text{z}=1} + w'_{\text{y}=1,\text{z}=0}) - \text{Pr}_{\text{train}}(\text{y} = 1)| \leq \gamma_\text{y}, \quad |(w'_{\text{y}=1,\text{z}=1} + w'_{\text{y}=0,\text{z}=1}) - \text{Pr}_{\text{train}}(\text{z} = 1)| \leq \gamma_\text{z},$$

$$\sum_{\forall y,z} w'_{\text{y}=y,\text{z}=z} = 1, \quad 0 \leq w'_{\text{y}=y,\text{z}=z} \leq 1, \quad \forall y \in \{0,1\}, z \in \{0,1\}$$

where $\text{Pr}_{\text{train}}(\text{y} = 1) = w_{\text{y}=1,\text{z}=1} + w_{\text{y}=1,\text{z}=0}$ and $\text{Pr}_{\text{train}}(\text{z} = 1) = w_{\text{y}=1,\text{z}=1} + w_{\text{y}=0,\text{z}=1}$. Note that when we know the exact shifted correlation value (i.e., $\alpha = \beta = c$), the first constraint can be rewritten as $w'_{\text{y}=1,\text{z}=1}/(w'_{\text{y}=1,\text{z}=1} + w'_{\text{y}=0,\text{z}=1}) - w'_{\text{y}=1,\text{z}=0}/(w'_{\text{y}=1,\text{z}=0} + w'_{\text{y}=0,\text{z}=0}) = c$.

The above optimization is a non-convex quadratically constrained quadratic problem (non-convex QCQP), where the objective is quadratic, and the first constraint is non-convex quadratic. However, the non-convex QCQP is known to be hard to solve (d'Aspremont & Boyd, 2003). Thus, we apply the semidefinite (SDP) relaxation, which is one of the convex relaxations known to give a reasonable lower bound of the optimal value of the original non-convex QCQP (Park & Boyd, 2017):

$$\min_{X,x} \quad \textbf{Tr}(XP_0) + q_0^T x$$

$$\text{s.t.} \quad \textbf{Tr}(XP_\alpha) \geq 0, \quad \textbf{Tr}(XP_\beta) \leq 0,$$

$$|q_2^T x - \text{Pr}_{\text{train}}(y = 1)| \leq \gamma_\text{y}, \quad |q_3^T x - \text{Pr}_{\text{train}}(z = 1)| \leq \gamma_\text{z}, \quad q_4^T x = 1, \quad 0 \leq x_i \leq 1 \; \forall i, \quad \begin{bmatrix} X & x \\ x^T & 1 \end{bmatrix} \succeq 0$$

where $\textbf{Tr}(\cdot)$ is the trace, $X = xx^T$, $x = \begin{bmatrix} x_1 & x_2 & x_3 & x_4 \end{bmatrix}^T = \begin{bmatrix} w'_{1,1} & w'_{1,0} & w'_{0,1} & w'_{0,0} \end{bmatrix}^T$, $q_0 = -2 \begin{bmatrix} w_{1,1} & w_{1,0} & w_{0,1} & w_{0,0} \end{bmatrix}^T$, $q_2 = \begin{bmatrix} 1 & 0 & 1 & 0 \end{bmatrix}^T$, $q_3 = \begin{bmatrix} 1 & 1 & 0 & 0 \end{bmatrix}^T$, $q_4 = \begin{bmatrix} 1 & 1 & 1 & 1 \end{bmatrix}^T$, $P_0 = \text{diag}(\mathbf{1})$, $P_\alpha = \begin{bmatrix} 0 & -\alpha/2 & 0 & (1-\alpha)/2 \\ -\alpha/2 & 0 & (-1-\alpha)/2 & 0 \\ 0 & (-1-\alpha)/2 & 0 & -\alpha/2 \\ (1-\alpha)/2 & 0 & -\alpha/2 & 0 \end{bmatrix}$, and $P_\beta = \begin{bmatrix} 0 & -\beta/2 & 0 & (1-\beta)/2 \\ -\beta/2 & 0 & (-1-\beta)/2 & 0 \\ 0 & (-1-\beta)/2 & 0 & -\beta/2 \\ (1-\beta)/2 & 0 & -\beta/2 & 0 \end{bmatrix}$.

Details on the conversion are in Sec. A.7. Since the above SDP relaxation problem is now convex, we can solve it using convex optimization solvers (e.g., CVXPY (Diamond & Boyd, 2016)).

## 4.2 OVERALL TRAINING

We present the overall process for fair training under correlation shifts in Algorithm 1. The algorithm includes two main parts: pre-processing for reflecting the shifted correlation and in-processing for fair training on top of the improved data. We first find the new (y, z)-class ratio $w_{\text{y},\text{z}}$ based on the SDP relaxation of our optimization, which can be solved using convex optimization solvers (e.g., CVXPY). We then calculate the sample-wise weights to ensure that the sample weight sum in each (y=$y$, z=$z$)-class is $w_{\text{y}=y,\text{z}=z} \cdot n$, where $n$ is the total number of samples in the original training data. Within each (y=$y$, z=$z$)-class, all samples have the same weight. We then

---

**Algorithm 1:** Fair Training under Correlation Shifts

**Input:** training data $D$, original ratio $w_{\text{y},\text{z}}$, correlation range $[\alpha, \beta]$, thresholds $\gamma_\text{y}$ and $\gamma_\text{z}$, in-processing algorithm $f$

$w'_{\text{y},\text{z}} = \texttt{SDPsolver}(w_{\text{y},\text{z}}, \alpha, \beta, \gamma_\text{y}, \gamma_\text{z})$

$d_j \leftarrow w'_{\text{y}=y,\text{z}=z}/w_{\text{y}=y,\text{z}=z}$,
$\quad \forall j \in \mathbb{I}_{(y,z)}, \forall (y,z) \in \mathbb{Y} \times \mathbb{Z}$

$\mathbf{d} = \{d_i\}_{i=1,\dots,n}$

(Optional) $\mathbf{d} = \texttt{MinDistChange}(D, w_{\text{y},\text{z}}, w'_{\text{y},\text{z}})$

Draw new data $D'$ from $D$ via weighted sampling w.r.t. $\mathbf{d}$

$\theta \leftarrow$ initial model parameters

**for** *each epoch* **do**: Update $\theta$ based on $f$ on $D'$

**Output :** $\theta$

---

draw new data $D'$ from the original training data via weighted sampling according to the sample-wise weights. Finally, we train a model using an in-processing fair algorithm $f$ on $D'$.

In addition, using an optional step (MinDistChange), we can address the scenario where the pre-processing should minimally change the original training data, which is sometimes preferred in other applications (Kamiran & Calders, 2011). Details on this extension are in Sec. A.8.

## 5 EXPERIMENTS

We perform experiments to evaluate our framework. We use logistic regression in all experiments. We evaluate the models on separate test datasets and repeat all experiments with five different random seeds. We use CVXPY as a convex optimization solver. More detailed settings are in Sec. B.3.

**Fairness metrics** We focus on two prominent group fairness metrics: demographic parity (DP) (Feldman et al., 2015) and equalized odds (EO) (Hardt et al., 2016). We measure the fairness disparities (i.e., unfairness) among sensitive groups as follows: *DP disparity* = $\max_{z \in \mathbb{Z}} |\Pr(\hat{y} = 1|z = z) - \Pr(\hat{y} = 1)|$ and *EO disparity* = $\max_{z \in \mathbb{Z}, y \in \mathbb{Y}} |\Pr(\hat{y} = y|z = z, y = y) - \Pr(\hat{y} = y|y = y)|$. When measuring the unfairness w.r.t. both DP and EO, we take the maximum of both disparities (i.e., $\max(DP\ disp., EO\ disp.)$). Note that lower disparity means better fairness.

**Datasets** We use a total of three datasets for training: one synthetic dataset and two real-world benchmark datasets. We generate the synthetic dataset using a method similar to Zafar et al. (2017a). The synthetic training dataset has 2,000 samples and consists of two non-sensitive attributes $(x_1, x_2)$, one sensitive attribute z, and one label attribute y – see details in Sec. B.3. We also utilize two real datasets: ProPublica COMPAS (Angwin et al., 2016) consists of 5,278 samples, and its labels indicate recidivism; AdultCensus (Kohavi, 1996) has 43,131 samples, and its labels indicate a person's annual income. We use gender as the sensitive attribute.

To construct test data representing the deployment distribution with shifted correlation ($c_{\text{test}}$), we use three methods: (1) re-sampling data within each (y, z)-class in the original test data, (2) modifying the z values while fixing the x and y distributions in the original test data (see details in Sec. B.3), and (3) utilizing a newly-collected data, where we train on the AdultCensus dataset (Kohavi, 1996), but test on a recent version of this dataset called ACSIncome (Ding et al., 2021).

**Baselines** We compare our framework with three types of baselines: (1) *vanilla (non-fair)* training using logistic regression, (2) *in-processing-only* training, and (3) *two-step* training that first runs an existing pre-processing algorithm and then an in-processing algorithm. For in-processing-only training, we use the following three approaches: Fairness Constraints (FC) (Zafar et al., 2017a;b), which adds an unfairness penalty term to the loss function; Adversarial Debiasing (AD) (Zhang et al., 2018), which adversarially trains a classifier with a fairness discriminator; and FairBatch (FB) (Roh et al., 2021), which adaptively adjusts batch ratios among groups to improve fairness. When we run in-processing approaches for multiple fairness metrics, we naturally extend each approach by combining the fairness constraints for different metrics – see details in Sec. B.3. For two-step training, we use a pre-processing algorithm called Reweighing (RW) (Kamiran & Calders, 2011) to debias the training data before running the above in-processing algorithms, where RW balances the data amounts across groups. We thus run the in-processing algorithms on the less-biased data by RW.

**Hyperparameters** We consider *three scenarios of knowing the range of the correlation constant* $c$ in the test data. As discussed in Sec. 4.1, one can infer this range using distribution estimation techniques (Huang et al., 2006; Zhang et al., 2013), which gives a confidence interval that can be used as the correlation constant range $[\alpha, \beta]$. **(1)** In Secs. 5.1 and 5.2, we assume that the range indicates the exact $c_{\text{test}}$ value (i.e., $\alpha = \beta = c_{\text{test}}$). **(2)** In Sec. B.10, we assume that the range is $c_{\text{test}} \pm x\%$. **(3)** In Sec. 5.3, we assume that the range is incorrect. We set both $\gamma_y$ and $\gamma_z$ to 0.1, which is a larger value than the actual marginal probability changes in the test data. We also test for the values of 0.2 and 0.3, and the overall trends remain the same. The in-processings' hyperparameters are in Sec. B.3.

### 5.1 ACCURACY AND FAIRNESS

We first compare the accuracy and fairness performances of our framework with baselines on the synthetic and COMPAS datasets in Table 1 – see many more results in Appendix B, including the AdultCensus experiments (Sec. B.4), which show similar results. The test data has lower correlation than the training data. LR in the first row shows vanilla training without any fairness technique. Other baselines are clustered based on three in-processing approaches: FC, AD, and FB. For each in-processing approach X, applying our pre-processing (denoted as Ours + X) generally shows better fairness while achieving comparable or even better accuracies, either when supporting only DP or both DP and EO. The in-processing-only baselines mostly show worse fairness and accuracies compared to applying our approach, because the in-processing-only baselines are trained with different biases from test data distribution in mind. The baselines of applying RW before in-processing generally do not achieve high fairness compared to ours. The reason is that existing pre-processing approaches like RW simply mitigate data bias as much as possible, which is not always beneficial for the in-processing. In comparison, our approach takes a more principled approach by adjusting the bias according to

Table 1: Performances on the synthetic and COMPAS test datasets w.r.t. a single metric (DP) and multiple metrics (DP & EO). The test datasets are constructed via re-sampling from the original distribution. The correlation constant $c$ of the test data is 50% of that of the training data. We compare our framework with three types of baselines: (1) non-fair training: LR; (2) in-processing-only training: FC, AD, and FB; (3) two-step training: RW (Kamiran & Calders, 2011) + in-processings. In the last row, we also show the performances of an in-processing algorithm (FB) trained on the test distribution, which can be considered as the upper bounds.

| | Synthetic | | | | COMPAS | | | |
| | Single (DP) | | Multiple (DP & EO) | | Single (DP) | | Multiple (DP & EO) | |
| Method | Acc. | Unfair. | Acc. | Unfair. | Acc. | Unfair. | Acc. | Unfair. |
|---|---|---|---|---|---|---|---|---|
| LR | $.865 \pm .000$ | $.173 \pm .000$ | $.865 \pm .000$ | $.173 \pm .000$ | $.660 \pm .000$ | $.129 \pm .000$ | $.660 \pm .000$ | $.225 \pm .000$ |
| FC (Zafar et al., 2017a;b) | $.778 \pm .011$ | $.038 \pm .013$ | $.853 \pm .001$ | $.075 \pm .002$ | $.656 \pm .004$ | $.050 \pm .021$ | $.654 \pm .007$ | $.137 \pm .034$ |
| RW+FC | $.848 \pm .004$ | $.079 \pm .003$ | $.848 \pm .004$ | $.082 \pm .002$ | $.654 \pm .004$ | $.068 \pm .039$ | $.651 \pm .010$ | $.124 \pm .031$ |
| **Ours**+FC | $.849 \pm .002$ | $\mathbf{.034 \pm .004}$ | $.851 \pm .005$ | $\mathbf{.060 \pm .003}$ | $.657 \pm .008$ | $\mathbf{.037 \pm .021}$ | $.652 \pm .020$ | $\mathbf{.106 \pm .038}$ |
| AD (Zhang et al., 2018) | $.762 \pm .016$ | $.032 \pm .011$ | $.821 \pm .004$ | $.068 \pm .003$ | $.655 \pm .003$ | $.054 \pm .015$ | $.661 \pm .005$ | $\mathbf{.111 \pm .035}$ |
| RW+AD | $.845 \pm .002$ | $.087 \pm .006$ | $.847 \pm .003$ | $.084 \pm .005$ | $.657 \pm .005$ | $.092 \pm .022$ | $.655 \pm .007$ | $.145 \pm .056$ |
| **Ours**+AD | $.814 \pm .011$ | $\mathbf{.017 \pm .006}$ | $.842 \pm .009$ | $\mathbf{.054 \pm .008}$ | $.650 \pm .003$ | $\mathbf{.045 \pm .008}$ | $.664 \pm .006$ | $.117 \pm .039$ |
| FB (Roh et al., 2021) | $.821 \pm .000$ | $.048 \pm .000$ | $.849 \pm .001$ | $.091 \pm .005$ | $.647 \pm .001$ | $.038 \pm .013$ | $.650 \pm .002$ | $.187 \pm .019$ |
| RW+FB | $.859 \pm .002$ | $.055 \pm .003$ | $.855 \pm .001$ | $.071 \pm .008$ | $.653 \pm .003$ | $.094 \pm .016$ | $.652 \pm .003$ | $.197 \pm .020$ |
| **Ours**+FB | $.836 \pm .001$ | $\mathbf{.003 \pm .001}$ | $.852 \pm .004$ | $\mathbf{.058 \pm .001}$ | $.648 \pm .004$ | $\mathbf{.027 \pm .001}$ | $.657 \pm .004$ | $\mathbf{.130 \pm .014}$ |
| *FB on test dist. (upper bound)* | $.838 \pm .002$ | $.003 \pm .002$ | $.859 \pm .002$ | $.058 \pm .004$ | $.659 \pm .001$ | $.012 \pm .008$ | $.664 \pm .001$ | $.095 \pm .014$ |

the correlation shift with the purpose of improving the in-processing performance. As a result, ours enables the in-processing approaches to be closer to optimal performances (last row). In Sec. B.5, we show that the pre-processed data by our algorithm is *more aligned* with the true test distribution than the original training data in terms of $(y, z)$-correlation and Wasserstein distance.

We observe similar results when using the *two other test settings* explained above. One is to modify the z values while fixing the x and y distributions using the synthetic dataset (Sec. B.6). The other is to use two income datasets collected in the 1990s (i.e., AdultCensus) and 2010s (i.e., ACSIncome) for training and testing, respectively, where they have different $(y, z)$-correlation values (Sec. B.7).

In addition, we enrich the experimental results by 1) presenting accuracy and fairness *trade-off curves* of the in-processing-only baseline and our approach (Sec. B.8) and 2) showing the effects of the optional step in Algorithm 1 (Sec. B.9), which finds possibly-different sample weights within each $(y, z)$-class to minimize the overall distribution change between the training and pre-processed data.

## 5.2 VARYING THE CORRELATION OF THE TEST DATA

We compare the algorithm performances when varying the correlation of the test data. Figure 3a shows the accuracy and fairness performances of FB and Ours+FB. As the test data's correlation constant $c$ varies from 10% to 70% of the training data's correlation constant, our pre-processing improves the accuracy and fairness of the in-processing algorithm. Interestingly, when the correlation of the test data differs significantly from the training data (i.e., close to 10%), the in-processing-only baseline (FB) shows worse fairness. We suspect that the baseline is mitigating a different type of bias than that of the test data. On the other hand, our pre-processing successfully enables the in-processing algorithm to achieve high fairness and accuracy (e.g., for a 10% test corr., the unfairness decreases from 0.108 to 0.003). In addition, we vary the correlation from 110% to 150% in Sec. B.11 where we show how our pre-processing enables in-processing to achieve high fairness.

## 5.3 HANDLING UNKNOWN CORRELATIONS

We also evaluate our approach when the exact shifted correlation is unknown with two scenarios: 1) misspecifying the correlation in the algorithm and 2) giving a range of correlation shifts to the algorithm. *[Scenario 1]* We first consider when the shifted correlation is *incorrectly specified*. We set $\alpha = \beta = c_{\text{specified}}$, where $c_{\text{specified}} \neq c_{\text{test}}$. Figure 3b shows the performances of FB and Ours+FB when the true correlation of the test data is 60% correlation of the training data. Ours improves the in-processing-only baseline's accuracy for the entire range of considered correlations and improves fairness when the specified correlation is higher than 30%. Hence, our approach is still beneficial when the estimation error is within 10%. *[Scenario 2]* In Sec. B.10, we also run our approach with *a range of correlation shifts*, which can be inferred by distribution estimation techniques (Huang et al., 2006; Zhang et al., 2013). We expand the $[\alpha, \beta]$ range to be $[c_{\text{test}} - x\%, c_{\text{test}} + x\%]$ instead of a constant value. As a result, there are two takeaways: 1) our framework successfully boosts the in-processing-only baseline performances when the $[\alpha, \beta]$ range is reasonable, and 2) even if we do not have any information about the correlation shift, our framework performs at least as well as the in-processing-only baselines.

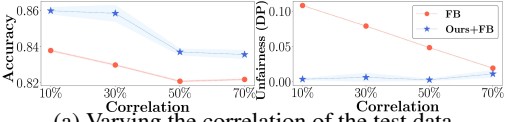 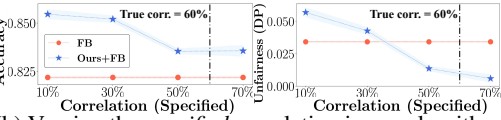

(a) Varying the correlation of the test data.      (b) Varying the *specified* correlation in our algorithm.

Figure 3: Performances of FB and Ours+FB on the synthetic data while (a) varying the correlation of the test data to have 10% to 70% correlation of the training data and (b) varying the *specified* correlation in our algorithm to have 10% to 70% where the true correlation of the test data is 60%.

## 6 RELATED WORK

As model fairness becomes essential for Trustworthy AI, various fairness techniques have been proposed to improve group fairness that do not discriminate specific demographics (Barocas et al., 2019) Usually, the fairness techniques can be categorized into three prominent approaches: pre-, in-, and post-processings – see their representative works in Sec. C.1. Among the three categories, in-processing approaches are widely used for fair training due to their high fairness and accuracy performances, but most of them assume that the training and deployment distributions are the same.

Recently, there is an emerging focus on fair training under data distribution shifts. In Table 2, we summarize various types of distribution shifts into the following two categories:

*General distribution shifts* (Singh et al., 2021; Rezaei et al., 2021; Chen et al., 2022; Mishler & Dalmasso, 2022) focus on shifts involving the input feature (x) and label (y), which are widely studied in the traditional machine learning literature. Here the bias changes between label (y) and group (z) are not explicitly considered.

Table 2: Different distribution shifts.

| Category | Type of Shifts | |
|---|---|---|
| General distribution shifts | covariate shift | $\Pr(x)$ |
| | label shift | $\Pr(y)$ |
| | concept shift | $\Pr(y\|x)$ |
| Fairness specific shifts | demographic shift | $\Pr(z)$ |
| | subpopulation shift | $\Pr(y, z)$ |
| | **correlation shift (ours)** | $\Pr(z\|y)$ |

*Fairness-specific shifts* (Maity et al., 2021; An et al., 2022; Giguere et al., 2022) handle group (z) distribution changes, as z is especially correlated with fair training. We note that our work also falls into this category. A recent study (Maity et al., 2021) theoretically analyzes the behavior of fair training under a change in bias called subpopulation shifts, where a specific group has fewer positively-labeled examples during training time compared to deployment time. Another study (Giguere et al., 2022) designs a new test method to serve a fair model under another data distribution change called demographic shifts, where the subgroup distribution may change – see an empirical comparison between this work and ours in Sec. B.12. A recent work (An et al., 2022) proposes a self-training-based transfer algorithm that requires specific model architecture (e.g., adversary network) to support data changes, including subpopulation shifts. In comparison, our contribution lies in 1) introducing the notion of correlation shifts, which is important for explaining with theoretical evidence the connection between the data bias changes and behaviors of fair training, 2) analyzing the fundamental performance limits of in-processing approaches in the presence of correlation shifts, and 3) proposing a pre-processing step based on the theoretical analysis for assisting the existing fairness approaches. In addition, our framework is general and can support any model architecture and training procedure. We leave more detailed comparisons in Sec. C.2.

Another line of research is supporting robustness in fair training, including handling noisy groups (Celis et al., 2021; Wang et al., 2020) or poisoning attacks (Mehrabi et al., 2021; Solans et al., 2020). Although this direction is not our immediate focus, we do perform preliminary experiments in Secs. B.13 and B.14 to show some potential to support noisy group attributes or poisoning attack scenarios. We believe our method can be further extended with other robust training methods.

In addition to group fairness, we discuss other noteworthy fairness definitions including individual fairness (Dwork et al., 2012a) and causality-based fairness (Kilbertus et al., 2017) in Sec. C.1. We also explain with a concrete example how our work connects to causality-based fairness in Sec. C.3.

## 7 CONCLUSION

We addressed the problem of model fairness in the presence of bias changes in the data. We first introduced the new notion of $(y, z)$-correlation for capturing bias and analyzed the accuracy and fairness limitations of existing in-processing approaches in the presence of correlation shifts. We then proposed a decoupling framework where pre-processing is used to adjust the correlation, and in-processing is used for unfairness mitigation. The pre-processing step adjusts the data ratio among $(y, z)$-classes to reflect the shifted correlation and can optionally minimize the distribution change of the training data as well. Experiments showed how our pre-processing enables existing in-processing approaches to achieve high fairness and accuracy under correlation shifts and outperform baselines.

ETHICS STATEMENT

We believe our work can have a positive societal impact by improving model fairness. In particular, we anticipate that our work can make existing fairness algorithms more applicable to real-world scenarios where data bias usually changes. However, one must carefully choose the right fairness metric to avoid unintended discrimination. To ensure privacy, we do not use any personal identifiers (e.g., name and date of birth). Also, we do not conduct experiments with human subjects.

REPRODUCIBILITY STATEMENT

To reproduce the experimental results, we describe details on experiments and implementation (e.g., models, data preprocessing, hyperparameters, and devices) in Sec. 5 and Sec. B. We also provide source codes for training and testing in the supplementary.

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

# A    APPENDIX – THEORY

## A.1    PROOF FOR LEMMA 2

Continuing from Sec. 3.1, we provide a proof for Lemma 2.

*Proof.* We denote the Pearson's correlation coefficient between y and z as $\rho_{yz}$. By definition, $\rho_{yz} = \frac{Cov(y,z)}{\sigma(y)\sigma(z)}$. For binary y and z, we can rewrite $\rho_{yz}$ as follows (Cohen & Cohen, 1975):

$$\rho_{yz} = \frac{Cov(y,z)}{\sigma(y)\sigma(z)} = \frac{\Pr(y=1,z=1)\Pr(y=0,z=0) - \Pr(y=1,z=0)\Pr(y=0,z=1)}{\sqrt{\Pr(y=1)\Pr(y=0)\Pr(z=1)\Pr(z=0)}}$$

If the marginal probabilities of y and z (i.e., $\Pr(y=y)$ and $\Pr(z=z)$) remain the same,

$$\begin{aligned}
\rho_{yz} &\propto \Pr(y=1,z=1)\Pr(y=0,z=0) - \Pr(y=1,z=0)\Pr(y=0,z=1) \\
&\propto \frac{\Pr(y=1,z=1)}{\Pr(z=1)}\frac{\Pr(y=0,z=0)}{\Pr(z=0)} - \frac{\Pr(y=1,z=0)}{\Pr(z=0)}\frac{\Pr(y=0,z=1)}{\Pr(z=1)} \\
&= \Pr(y=1|z=1)\Pr(y=0|z=0) - \Pr(y=1|z=0)\Pr(y=0|z=1) \\
&= \Pr(y=1|z=1)(1-\Pr(y=1|z=0)) - \Pr(y=1|z=0)(1-\Pr(y=1|z=1)) \\
&= \Pr(y=1|z=1) - \Pr(y=1|z=0) - \Pr(y=1|z=1)\Pr(y=1|z=0) + \Pr(y=1|z=0)\Pr(y=1|z=1) \\
&= \Pr(y=1|z=1) - \Pr(y=1|z=0).
\end{aligned}$$

Therefore, if the marginal probabilities of y and z remain the same, the $(y,z)$-correlation $\rho_{yz}$ is proportional to $\Pr(y=1|z=1) - \Pr(y=1|z=0)$, which is the difference between the conditional probabilities of y given different z values.

□

## A.2    THE ACCURACY-FAIRNESS TRADEOFF WHEN IMPROVING FAIRNESS W.R.T. A SINGLE METRIC

Continuing from Sec. 3.1, we show that the $(y,z)$-correlation determines the accuracy-fairness tradeoff under certain conditions by applying Lemma 2 to Proposition 1.

The previous work (Menon & Williamson, 2018) shows that a higher alignment between y and z leads to a worse accuracy-fairness tradeoff w.r.t. demographic parity – more details are described in Proposition 8 in Menon & Williamson (2018). According to the previous work, alignment is defined as how many examples in each group have a specific label. We can thus measure the alignment as $\Pr(y=1|z=1) + \Pr(y=0|z=0)$. Then, we can rewrite the equation as follows:

$$\begin{aligned}
\text{alignment} &= \Pr(y=1|z=1) + \Pr(y=0|z=0) \\
&= \Pr(y=1|z=1) + (1 - \Pr(y=1|z=0)) \\
&= 1 + \Pr(y=1|z=1) - \Pr(y=1|z=0).
\end{aligned}$$

By applying Lemma 2 to the above result, we observe that the alignment between y and z is proportional to $((y,z)$-correlation $+1)$ when the marginal distributions of y and z remain the same. Therefore, when a model is trained w.r.t. demographic parity and the marginal probabilities of y and z do not change, a higher $(y,z)$-correlation results in a worse accuracy-fairness tradeoff.

## A.3    $\varepsilon$-DP & $\varepsilon$-EO

Continuing from Sec. 3.1, we provide a proof for Proposition 6.

*Proof.* A model achieves both $\varepsilon$-DP and $\varepsilon$-EO when the two inequalities $|\Pr(\hat{y}=y|z=z) - \Pr(\hat{y}=y|z=z')| \leq \varepsilon$ (i.e., $\varepsilon$-DP) and $|\Pr(\hat{y}=y|y=y,z=z) - \Pr(\hat{y}=y|y=y,z=z')| \leq \varepsilon$ (i.e., $\varepsilon$-EO) are satisfied, where $y \neq y', y, y' \in \{0,1\}$ and $z \neq z', z, z' \in \{0,1\}$.

By combining the law of total probability and $\varepsilon$-DP, we can get the following inequality:

$$
\begin{aligned}
-\varepsilon \leq &\Pr(\hat{y} = y | z = z, y = y') \Pr(y = y' | z = z) + \Pr(\hat{y} = y | z = z, y = y) \Pr(y = y | z = z) \\
&- \Pr(\hat{y} = y | z = z', y = y') \Pr(y = y' | z = z') - \Pr(\hat{y} = y | z = z', y = y) \Pr(y = y | z = z') \leq \varepsilon.
\end{aligned}
\tag{1}
$$

Also, $\varepsilon$-EO can be rewritten as follows:

$$
-\varepsilon + \Pr(\hat{y} = y | z = z', y = y) \leq \Pr(\hat{y} = y | z = z, y = y) \leq \varepsilon + \Pr(\hat{y} = y | z = z', y = y). \tag{2}
$$

By substituting Eq. 2 to Eq. 1, we can get the following inequality:

$$
\begin{aligned}
&(-\varepsilon + \Pr(\hat{y} = y | z = z', y = y')) \Pr(y = y' | z = z) + (-\varepsilon + \Pr(\hat{y} = y | z = z', y = y)) \Pr(y = y | z = z) \\
&- \Pr(\hat{y} = y | z = z', y = y') \Pr(y = y' | z = z') - \Pr(\hat{y} = y | z = z', y = y) \Pr(y = y | z = z') \\
&\leq \Pr(\hat{y} = y | z = z, y = y') \Pr(y = y' | z = z) + \Pr(\hat{y} = y | z = z, y = y) \Pr(y = y | z = z) \qquad (3) \\
&- \Pr(\hat{y} = y | z = z', y = y') \Pr(y = y' | z = z') - \Pr(\hat{y} = y | z = z', y = y) \Pr(y = y | z = z') \\
&\leq (\varepsilon + \Pr(\hat{y} = y | z = z', y = y')) \Pr(y = y' | z = z) + (\varepsilon + \Pr(\hat{y} = y | z = z', y = y)) \Pr(y = y | z = z) \\
&- \Pr(\hat{y} = y | z = z', y = y') \Pr(y = y' | z = z') - \Pr(\hat{y} = y | z = z', y = y) \Pr(y = y | z = z').
\end{aligned}
$$

By subtracting Eq. 2 from Eq. 3,

$$
\begin{aligned}
&-\varepsilon + (-\varepsilon + \Pr(\hat{y} = y | z = z', y = y')) \Pr(y = y' | z = z) + (-\varepsilon + \Pr(\hat{y} = y | z = z', y = y)) \Pr(y = y | z = z) \\
&- \Pr(\hat{y} = y | z = z', y = y') \Pr(y = y' | z = z') - \Pr(\hat{y} = y | z = z', y = y) \Pr(y = y | z = z') \\
&\leq 0 \\
&\leq \varepsilon + (\varepsilon + \Pr(\hat{y} = y | z = z', y = y')) \Pr(y = y' | z = z) + (\varepsilon + \Pr(\hat{y} = y | z = z', y = y)) \Pr(y = y | z = z) \\
&- \Pr(\hat{y} = y | z = z', y = y') \Pr(y = y' | z = z') - \Pr(\hat{y} = y | z = z', y = y) \Pr(y = y | z = z').
\end{aligned}
$$

By rearranging the terms, we get the followings:

$$
\begin{aligned}
-2\varepsilon \leq &(\Pr(\hat{y} = y | z = z', y = y') - \Pr(\hat{y} = y | z = z', y = y))(\Pr(y = y' | z = z) - \Pr(y = y' | z = z')) \leq 2\varepsilon \\
\Rightarrow -2\varepsilon \leq &(\Pr(\hat{y} = y | z = z', y = y') - \Pr(\hat{y} = y | z = z', y = y))(1 - \Pr(y = y | z = z) - 1 + \Pr(y = y | z = z')) \leq 2\varepsilon.
\end{aligned}
$$

Thus, we get the following inequality, which is in Proposition 6:

$$
|\Pr(y = y | z = z) - \Pr(y = y | z = z')| \, |\Pr(\hat{y} = y | y = y, z = z) - \Pr(\hat{y} = y | y = y', z = z)| \leq 2\varepsilon.
$$

$$\square$$

## A.4 $\varepsilon$-PP & $\varepsilon$-DP

Continuing from Sec. 3.1, we consider fair training w.r.t. both predictive parity (PP) and demographic parity (DP). We can define $\varepsilon$-PP & $\varepsilon$-DP as follows:

$$
\begin{aligned}
-\varepsilon \leq \Pr(y = y | \hat{y} = y, z = 0) - \Pr(y = y | \hat{y} = y, z = 1) \leq \varepsilon \quad (\text{i.e., } \varepsilon\text{-PP}) \\
-\varepsilon \leq \Pr(\hat{y} = y | z = 0) - \Pr(\hat{y} = y | z = 1) \leq \varepsilon \quad (\text{i.e., } \varepsilon\text{-DP})
\end{aligned}
$$

Based on these definitions, we give a proposition for $\varepsilon$-PP and $\varepsilon$-DP:

**Proposition 8** ($\varepsilon$-PP & $\varepsilon$-DP). *Let a model achieve both $\varepsilon$-PP and $\varepsilon$-DP. Then, the following inequality holds:*

$$
\frac{|\Pr(y = y, \hat{y} = y | z = z) - \Pr(y = y, \hat{y} = y | z = z')|}{2 \Pr(y = y | \hat{y} = y, z = z) + \Pr(\hat{y} = y | z = z) + \Pr(y = y | \hat{y} = y, z = z')} \leq \varepsilon, \ \ z \neq z', \ z, z' \in \mathbb{Z}, \ y \in \mathbb{Y}.
$$

*Proof.* From $\varepsilon$-DP, we get

$$
-\varepsilon + \Pr(\hat{y} = y | z = 1) \leq \Pr(\hat{y} = y | z = 0) \leq \varepsilon + \Pr(\hat{y} = y | z = 1). \tag{4}
$$

From $\varepsilon$-PP, we get

$$-\varepsilon \le \Pr(\hat{y}{=}y|z{=}0)\frac{\Pr(z{=}0)}{\Pr(\hat{y}{=}y,z{=}0)}\frac{\Pr(y{=}y,\hat{y}{=}y,z{=}0)}{\Pr(\hat{y}{=}y,z{=}0)} - \Pr(\hat{y}{=}y|z{=}1)\frac{\Pr(z{=}1)}{\Pr(\hat{y}{=}y,z{=}1)}\frac{\Pr(y{=}y,\hat{y}{=}y,z{=}1)}{\Pr(\hat{y}{=}y,z{=}1)} \le \varepsilon.$$
(5)

By substituting Eq. 4 to Eq. 5, we can get the following inequality:

$$\{-\varepsilon + \Pr(\hat{y}=y|z=1)\}\frac{\Pr(z=0)}{\Pr(\hat{y}=y,z=0)}\frac{\Pr(y=y,\hat{y}=y,z=0)}{\Pr(\hat{y}=y,z=0)} - \Pr(\hat{y}=y|z=1)\frac{\Pr(z=1)}{\Pr(\hat{y}=y,z=1)}\frac{\Pr(y=y,\hat{y}=y,z=1)}{\Pr(\hat{y}=y,z=1)}$$
$$\le \Pr(\hat{y}=y|z=0)\frac{\Pr(z=0)}{\Pr(\hat{y}=y,z=0)}\frac{\Pr(y=y,\hat{y}=y,z=0)}{\Pr(\hat{y}=y,z=0)} - \Pr(\hat{y}=y|z=1)\frac{\Pr(z=1)}{\Pr(\hat{y}=y,z=1)}\frac{\Pr(y=y,\hat{y}=y,z=1)}{\Pr(\hat{y}=y,z=1)}$$
(6)
$$\le \{\varepsilon + \Pr(\hat{y}=y|z=1)\}\frac{\Pr(z=0)}{\Pr(\hat{y}=y,z=0)}\frac{\Pr(y=y,\hat{y}=y,z=0)}{\Pr(\hat{y}=y,z=0)} - \Pr(\hat{y}=y|z=1)\frac{\Pr(z=1)}{\Pr(\hat{y}=y,z=1)}\frac{\Pr(y=y,\hat{y}=y,z=1)}{\Pr(\hat{y}=y,z=1)}.$$

By subtracting Eq. 5 from Eq. 6,

$$-\varepsilon + \{-\varepsilon + \Pr(\hat{y}{=}y|z{=}1)\}\frac{\Pr(z=0)}{\Pr(\hat{y}=y,z=0)}\frac{\Pr(y=y,\hat{y}=y,z=0)}{\Pr(\hat{y}=y,z=0)} - \Pr(\hat{y}{=}y|z{=}1)\frac{\Pr(z=1)}{\Pr(\hat{y}=y,z=1)}\frac{\Pr(y=y,\hat{y}=y,z=1)}{\Pr(\hat{y}=y,z=1)}$$
$$\le 0$$
$$\le \varepsilon + \{\varepsilon + \Pr(\hat{y}{=}y|z{=}1)\}\frac{\Pr(z=0)}{\Pr(\hat{y}=y,z=0)}\frac{\Pr(y=y,\hat{y}=y,z=0)}{\Pr(\hat{y}=y,z=0)} - \Pr(\hat{y}{=}y|z{=}1)\frac{\Pr(z=1)}{\Pr(\hat{y}=y,z=1)}\frac{\Pr(y=y,\hat{y}=y,z=1)}{\Pr(\hat{y}=y,z=1)}.$$

By rearranging the terms, we get the following inequality:

$$-\varepsilon \cdot \frac{\Pr(z=0)\Pr(y=y,\hat{y}=y,z=0)}{(\Pr(\hat{y}=y,z=0))^2} - \varepsilon$$
$$\le \Pr(\hat{y}=y|z=1)\{\frac{\Pr(z=1)\Pr(y=y,\hat{y}=y,z=1)}{(\Pr(\hat{y}=y,z=1))^2} - \frac{\Pr(z=0)\Pr(y=y,\hat{y}=y,z=0)}{(\Pr(\hat{y}=y,z=0))^2}\}$$
$$\le \varepsilon \cdot \frac{\Pr(z=0)\Pr(y=y,\hat{y}=y,z=0)}{(\Pr(\hat{y}=y,z=0))^2} + \varepsilon$$

which is rewritten as follows:

$$-\varepsilon \cdot \frac{\Pr(y=y|\hat{y}=y,z=0) + \Pr(\hat{y}=y|z=0)}{\Pr(\hat{y}=y|z=0)}$$
$$\le \Pr(\hat{y}=y|z=1)\{\frac{\Pr(y=y|\hat{y}=y,z=1)}{\Pr(\hat{y}=y|z=1)} - \frac{\Pr(y=y|\hat{y}=y,z=0)}{\Pr(\hat{y}=y|z=0)}\}$$
$$\le \varepsilon \cdot \frac{\Pr(y=y|\hat{y}=y,z=0) + \Pr(\hat{y}=y|z=0)}{\Pr(\hat{y}=y|z=0)}.$$

By multiplying $\frac{\Pr(\hat{y}=y|z=0)}{\Pr(y=y|\hat{y}=y,z=0)+\Pr(\hat{y}=y|z=0)}$ for all terms in the above inequality, we can get

$$-\varepsilon \le \frac{\Pr(\hat{y}=y|z=0)\Pr(y=y|\hat{y}=y,z=1) - \Pr(\hat{y}=y|z=1)\Pr(y=y|\hat{y}=y,z=0)}{\Pr(y=y|\hat{y}=y,z=0) + \Pr(\hat{y}=y|z=0)} \le \varepsilon.$$
(7)

Let $A = \Pr(y=y|\hat{y}=y,z=0) + \Pr(\hat{y}=y|z=0)$. Since $-\varepsilon \le \Pr(\hat{y}=y|z=0) - \Pr(\hat{y}=y|z=1) \le \varepsilon$ (i.e., $\varepsilon$-DP), we can make another inequality from Eq. 7:

$$\frac{(-\varepsilon + \Pr(\hat{y}=y|z=1))\Pr(y=y|\hat{y}=y,z=1) - (\varepsilon + \Pr(\hat{y}=y|z=0))\Pr(y=y|\hat{y}=y,z=0)}{A}$$
$$\le \frac{\Pr(\hat{y}=y|z=0)\Pr(y=y|\hat{y}=y,z=1) - \Pr(\hat{y}=y|z=1)\Pr(y=y|\hat{y}=y,z=0)}{A}$$
(8)
$$\le \frac{(\varepsilon + \Pr(\hat{y}=y|z=1))\Pr(y=y|\hat{y}=y,z=1) - (-\varepsilon + \Pr(\hat{y}=y|z=0))\Pr(y=y|\hat{y}=y,z=0)}{A}.$$

By subtracting Eq. 7 and Eq. 8,

$$-\varepsilon + \frac{(-\varepsilon + \Pr(\hat{y} = y | z = 1)) \Pr(y = y | \hat{y} = y, z = 1) - (\varepsilon + \Pr(\hat{y} = y | z = 0)) \Pr(y = y | \hat{y} = y, z = 0)}{A}$$
$$\leq 0$$
$$\leq \varepsilon + \frac{(\varepsilon + \Pr(\hat{y} = y | z = 1)) \Pr(y = y | \hat{y} = y, z = 1) - (-\varepsilon + \Pr(\hat{y} = y | z = 0)) \Pr(y = y | \hat{y} = y, z = 0)}{A}.$$

Since $\Pr(\hat{y} = y | z = z) \Pr(y = y | \hat{y} = y, z = z) = \Pr(y = y, \hat{y} = y | z = z)$, we can rewritten the above inequality as follows:

$$-\varepsilon + \frac{-\varepsilon \{\Pr(y = y | \hat{y} = y, z = 1) + \Pr(y = y | \hat{y} = y, z = 0)\} + \Pr(y = y, \hat{y} = y | z = 1) - \Pr(y = y, \hat{y} = y | z = 0)}{A}$$
$$\leq 0$$
$$\leq \varepsilon + \frac{\varepsilon \{\Pr(y = y | \hat{y} = y, z = 1) + \Pr(y = y | \hat{y} = y, z = 0)\} + \Pr(y = y, \hat{y} = y | z = 1) - \Pr(y = y, \hat{y} = y | z = 0)}{A}.$$

Now, let $B = \Pr(y = y | \hat{y} = y, z = 1) + \Pr(y = y | \hat{y} = y, z = 0)$ and $C = \Pr(y = y, \hat{y} = y | z = 1) - \Pr(y = y, \hat{y} = y | z = 0)$. Then,

$$-\varepsilon + \frac{1}{A}(-\varepsilon \cdot B + C) \leq 0 \leq \varepsilon + \frac{1}{A}(\varepsilon \cdot B + C).$$

By arranging the terms,

$$-\varepsilon \cdot (1 + \frac{B}{A}) + \frac{C}{A} \leq 0 \leq \varepsilon \cdot (1 + \frac{B}{A}) + \frac{C}{A}$$
$$\implies -\varepsilon \leq \frac{-C}{A + B} \leq \varepsilon.$$

Therefore, we can conclude

$$\frac{|\Pr(y = y, \hat{y} = y | z = 0) - \Pr(y = y, \hat{y} = y | z = 1)|}{2 \Pr(y = y | \hat{y} = y, z = 0) + \Pr(\hat{y} = y | z = 0) + \Pr(y = y | \hat{y} = y, z = 1)} \leq \varepsilon.$$

If we reverse the z values in the derivation, we get the same formula with only the z value changed in the above expression. As a result, we get the following inequality, which is in Proposition 8:

$$\frac{|\Pr(y = y, \hat{y} = y | z = z) - \Pr(y = y, \hat{y} = y | z = z')|}{2 \Pr(y = y | \hat{y} = y, z = z) + \Pr(\hat{y} = y | z = z) + \Pr(y = y | \hat{y} = y, z = z')} \leq \varepsilon, \ z \neq z', \ z, z' \in \mathbb{Z}, \ y \in \mathbb{Y}.$$

$$\square$$

Therefore, to make $\varepsilon = 0$, the numerator term $|\Pr(y = y, \hat{y} = y | z = z) - \Pr(y = y, \hat{y} = y | z = z')|$ should be zero. When $|\Pr(y = y, \hat{y} = y | z = z) - \Pr(y = y, \hat{y} = y | z = z')| = 0$, the following is satisfied: $(y, \hat{y}) \perp z$, which implies $y \perp z$ and $\hat{y} \perp z$. Here, $y \perp z$ indicates that $|\Pr(y = y | z = z) - \Pr(y = y | z = z')|$ is zero. As a result, perfectly satisfying PP and DP requires the data to be fully unbiased. We thus suspect that the achievable model fairness w.r.t. both PP and DP is affected by the $(y, z)$-correlation.

## A.5 $\varepsilon$-EO & $\varepsilon$-PP

Continuing from Sec. 3.1, we consider fair training w.r.t. both equalized odds (EO) and predictive parity (PP). We can define $\varepsilon$-EO & $\varepsilon$-PP as follows:

$$-\varepsilon \leq \Pr(\hat{y} = y | y = y, z = 0) - \Pr(\hat{y} = y | y = y, z = 1) \leq \varepsilon \ \text{(i.e., } \varepsilon\text{-EO)}$$
$$-\varepsilon \leq \Pr(y = y | \hat{y} = y, z = 0) - \Pr(y = y | \hat{y} = y, z = 1) \leq \varepsilon \ \text{(i.e., } \varepsilon\text{-PP)}$$

Based on these definitions, we give a proposition for $\varepsilon$-EO and $\varepsilon$-PP:

**Proposition 9** ($\varepsilon$-EO & $\varepsilon$-PP)**.** *Let a model achieve both $\varepsilon$-EO and $\varepsilon$-PP. Then, the following inequality holds:*

$$\frac{\Pr(\hat{y}{=}y|\mathbf{y}{=}y, \mathbf{z}{=}z')}{\Pr(\hat{y}{=}y, \mathbf{z}{=}z) + \Pr(\mathbf{y}{=}y, \mathbf{z}{=}z)} \cdot |\Pr(\mathbf{y}{=}y) - \Pr(\mathbf{y}{=}y|\mathbf{z}{=}z') \cdot \frac{\Pr(\hat{y}{=}y)}{\Pr(\hat{y}{=}y|\mathbf{z}{=}z')}| \leq \varepsilon, \ \ z \neq z', \ z, z' \in \mathbb{Z}, \ y \in \mathbb{Y}.$$

*Proof.* From $\varepsilon$-EO, we get

$$-\varepsilon + \Pr(\hat{y} = y|\mathbf{y} = y, \mathbf{z} = 1) \leq \Pr(\hat{y} = y|\mathbf{y} = y, \mathbf{z} = 0) \leq \varepsilon + \Pr(\hat{y} = y|\mathbf{y} = y, \mathbf{z} = 1). \quad (9)$$

From $\varepsilon$-PP, we get

$$-\varepsilon \leq \Pr(\hat{y} = y|\mathbf{y} = y, \mathbf{z} = 0)\frac{\Pr(\mathbf{y} = y, \mathbf{z} = 0)}{\Pr(\hat{y} = y, \mathbf{z} = 0)} - \Pr(\hat{y} = y|\mathbf{y} = y, \mathbf{z} = 1)\frac{\Pr(\mathbf{y} = y, \mathbf{z} = 1)}{\Pr(\hat{y} = y, \mathbf{z} = 1)} \leq \varepsilon. \quad (10)$$

By substituting Eq. 9 to Eq. 10, we can get the following inequality:

$$(-\varepsilon + \Pr(\hat{y} = y|\mathbf{y} = y, \mathbf{z} = 1))\frac{\Pr(\mathbf{y} = y, \mathbf{z} = 0)}{\Pr(\hat{y} = y, \mathbf{z} = 0)} - \Pr(\hat{y} = y|\mathbf{y} = y, \mathbf{z} = 1)\frac{\Pr(\mathbf{y} = y, \mathbf{z} = 1)}{\Pr(\hat{y} = y, \mathbf{z} = 1)}$$

$$\leq \Pr(\hat{y} = y|\mathbf{y} = y, \mathbf{z} = 0)\frac{\Pr(\mathbf{y} = y, \mathbf{z} = 0)}{\Pr(\hat{y} = y, \mathbf{z} = 0)} - \Pr(\hat{y} = y|\mathbf{y} = y, \mathbf{z} = 1)\frac{\Pr(\mathbf{y} = y, \mathbf{z} = 1)}{\Pr(\hat{y} = y, \mathbf{z} = 1)} \quad (11)$$

$$\leq (\varepsilon + \Pr(\hat{y} = y|\mathbf{y} = y, \mathbf{z} = 1))\frac{\Pr(\mathbf{y} = y, \mathbf{z} = 0)}{\Pr(\hat{y} = y, \mathbf{z} = 0)} - \Pr(\hat{y} = y|\mathbf{y} = y, \mathbf{z} = 1)\frac{\Pr(\mathbf{y} = y, \mathbf{z} = 1)}{\Pr(\hat{y} = y, \mathbf{z} = 1)}.$$

By subtracting Eq. 10 from Eq. 11,

$$-\varepsilon + (-\varepsilon + \Pr(\hat{y} = y|\mathbf{y} = y, \mathbf{z} = 1))\frac{\Pr(\mathbf{y} = y, \mathbf{z} = 0)}{\Pr(\hat{y} = y, \mathbf{z} = 0)} - \Pr(\hat{y} = y|\mathbf{y} = y, \mathbf{z} = 1)\frac{\Pr(\mathbf{y} = y, \mathbf{z} = 1)}{\Pr(\hat{y} = y, \mathbf{z} = 1)}$$

$$\leq 0$$

$$\leq \varepsilon + (\varepsilon + \Pr(\hat{y} = y|\mathbf{y} = y, \mathbf{z} = 1))\frac{\Pr(\mathbf{y} = y, \mathbf{z} = 0)}{\Pr(\hat{y} = y, \mathbf{z} = 0)} - \Pr(\hat{y} = y|\mathbf{y} = y, \mathbf{z} = 1)\frac{\Pr(\mathbf{y} = y, \mathbf{z} = 1)}{\Pr(\hat{y} = y, \mathbf{z} = 1)}.$$

By rearranging the terms, we get the following inequality:

$$-\varepsilon \cdot \frac{\Pr(\hat{y} = y, \mathbf{z} = 0) + \Pr(\mathbf{y} = y, \mathbf{z} = 0)}{\Pr(\hat{y} = y, \mathbf{z} = 0)}$$

$$\leq \Pr(\hat{y} = y|\mathbf{y} = y, \mathbf{z} = 1)\{\frac{\Pr(\mathbf{y} = y, \mathbf{z} = 0)}{\Pr(\hat{y} = y, \mathbf{z} = 0)} - \frac{\Pr(\mathbf{y} = y, \mathbf{z} = 1)}{\Pr(\hat{y} = y, \mathbf{z} = 1)}\}$$

$$\leq \varepsilon \cdot \frac{\Pr(\hat{y} = y, \mathbf{z} = 0) + \Pr(\mathbf{y} = y, \mathbf{z} = 0)}{\Pr(\hat{y} = y, \mathbf{z} = 0)}.$$

By multiplying $\frac{\Pr(\hat{y}=y, \mathbf{z}=0)}{\Pr(\hat{y}=y, \mathbf{z}=0) + \Pr(\mathbf{y}=y, \mathbf{z}=0)}$ for all terms in the inequality, we can get

$$-\varepsilon \leq \frac{\Pr(\hat{y} = y, \mathbf{z} = 0)\Pr(\hat{y} = y|\mathbf{y} = y, \mathbf{z} = 1)}{\Pr(\hat{y} = y, \mathbf{z} = 0) + \Pr(\mathbf{y} = y, \mathbf{z} = 0)}\{\frac{\Pr(\mathbf{y} = y, \mathbf{z} = 0)}{\Pr(\hat{y} = y, \mathbf{z} = 0)} - \frac{\Pr(\mathbf{y} = y, \mathbf{z} = 1)}{\Pr(\hat{y} = y, \mathbf{z} = 1)}\} \leq \varepsilon$$

which is rewritten as follows:

$$-\varepsilon \leq \frac{\Pr(\hat{y} = y|\mathbf{y} = y, \mathbf{z} = 1)}{\Pr(\hat{y} = y, \mathbf{z} = 0) + \Pr(\mathbf{y} = y, \mathbf{z} = 0)}\{\Pr(\mathbf{y} = y, \mathbf{z} = 0) - \frac{\Pr(\hat{y} = y, \mathbf{z} = 0)}{\Pr(\hat{y} = y, \mathbf{z} = 1)} \cdot \Pr(\mathbf{y} = y, \mathbf{z} = 1)\} \leq \varepsilon.$$

Since $\Pr(\mathbf{y} = y, \mathbf{z} = 0) = \Pr(\mathbf{y} = y) - \Pr(\mathbf{y} = y, \mathbf{z} = 1)$ by the total probability law,

$$-\varepsilon \leq \frac{\Pr(\hat{y}{=}y|\mathbf{y}{=}y, \mathbf{z}{=}1)}{\Pr(\hat{y}{=}y, \mathbf{z}{=}0) + \Pr(\mathbf{y}{=}y, \mathbf{z}{=}0)}\{\Pr(\mathbf{y}{=}y) - \Pr(\mathbf{y}{=}y, \mathbf{z}{=}1) - \frac{\Pr(\hat{y}{=}y, \mathbf{z}{=}0)}{\Pr(\hat{y}{=}y, \mathbf{z}{=}1)} \cdot \Pr(\mathbf{y}{=}y, \mathbf{z}{=}1)\} \leq \varepsilon.$$

By arranging the terms,

$$-\varepsilon \le \frac{\Pr(\hat{y} = y | y = y, z = 1)}{\Pr(\hat{y} = y, z = 0) + \Pr(y = y, z = 0)} \{\Pr(y = y) - \Pr(y = y, z = 1) \cdot (1 + \frac{\Pr(\hat{y} = y, z = 0)}{\Pr(\hat{y} = y, z = 1)}))\} \le \varepsilon$$

$$\Rightarrow -\varepsilon \le \frac{\Pr(\hat{y} = y | y = y, z = 1)}{\Pr(\hat{y} = y, z = 0) + \Pr(y = y, z = 0)} \{\Pr(y = y) - \Pr(y = y, z = 1) \cdot \frac{\Pr(\hat{y} = y)}{\Pr(\hat{y} = y, z = 1)}\} \le \varepsilon.$$

Therefore, we can conclude

$$\frac{\Pr(\hat{y} = y | y = y, z = 1)}{\Pr(\hat{y} = y, z = 0) + \Pr(y = y, z = 0)} \cdot |\Pr(y = y) - \Pr(y = y, z = 1) \cdot \frac{\Pr(\hat{y} = y)}{\Pr(\hat{y} = y, z = 1)}| \le \varepsilon.$$

If we reverse the z values in the derivation, we get the same formula with only the z value changed in the above expression. As a result, we get the following inequality, which is in Proposition 9:

$$\frac{\Pr(\hat{y}=y | y=y, z=z')}{\Pr(\hat{y}=y, z=z) + \Pr(y=y, z=z)} \cdot |\Pr(y=y) - \Pr(y=y | z=z') \cdot \frac{\Pr(\hat{y}=y)}{\Pr(\hat{y}=y | z=z')}| \le \varepsilon, \ z \ne z', \ z, z' \in \mathbb{Z}, \ y \in \mathbb{Y}.$$

$\square$

Therefore, $\varepsilon = 0$ when $\Pr(y = y) = \Pr(y = y | z = z')$ and $\Pr(\hat{y} = y) = \Pr(\hat{y} = y | z = z')$ (i.e., $y \perp z$ and $\hat{y} \perp z$). Here, satisfying both $y \perp z$ and $\hat{y} \perp z$ is the sufficient condition of perfectly satisfying EO and PP. Note that $y \perp z$ implies that the data is unbiased. We thus suspect that the achievable model fairness w.r.t. both EO and PP is affected by the $(y, z)$-correlation.

## A.6 Assumption on the distribution of input feature x

Continuing from Sec. 4, we clarify how we consider the input feature x: our framework (Sec. 4) implicitly assumes the distribution of x does not change drastically, but we empirically observe that our framework performs well in the real-world scenario when the x distribution changes (Sec. 5).

Our framework is designed to work best when the x distribution does not change, but does not strictly require this condition unlike other previous works on fairness-specific shifts (Maity et al., 2021; Giguere et al., 2022). Specifically, Algorithm 1 uses random sampling w.r.t. the new weights on each $(y, z)$-class, so it is desirable that the x distribution conditioned on each $(y, z)$-class stays the same. Of course, if the x distribution completely changes, the fairness or accuracy will degrade.

In this paper, we empirically show that our framework indeed performs well in real-world x distribution shift scenarios. For example, Table 6 in Section B.7 shows the accuracy and fairness performances when using the two income datasets collected in the 1990s (Kohavi, 1996) (i.e., AdultCensus dataset) and 2010s (Ding et al., 2021) (i.e., ACSIncome dataset) for training and testing, respectively. Since the two datasets are collected separately in different periods, both the $(y, z)$-correlation and the x distribution can be different between these datasets. Nevertheless, we observe that our framework successfully improves the accuracy and fairness performances of the in-processing-only baselines.

## A.7 Semidefinite Relaxation

Continuing from Sec. 4.1, we provide details of the semidefinite relaxation in our optimization.

Recall our original optimization as follows:

$$\min_{w'} \sum_{\forall y, z} (w_{y=y, z=z} - w'_{y=y, z=z})^2$$

$$\text{s.t. } \alpha \le \frac{w'_{y=1, z=1}}{w'_{y=1, z=1} + w'_{y=0, z=1}} - \frac{w'_{y=1, z=0}}{w'_{y=1, z=0} + w'_{y=0, z=0}} \le \beta,$$

$$|(w'_{y=1, z=1} + w'_{y=1, z=0}) - \Pr_{\text{train}}(y = 1)| \le \gamma_y,$$

$$|(w'_{y=1, z=1} + w'_{y=0, z=1}) - \Pr_{\text{train}}(z = 1)| \le \gamma_z,$$

$$\sum_{\forall y, z} w'_{y=y, z=z} = 1, \ 0 \le w'_{y=y, z=z} \le 1, \ \forall y \in \{0, 1\}, z \in \{0, 1\}$$

where $\Pr_{\text{train}}(\text{y} = 1) = w_{\text{y}=1,\text{z}=1} + w_{\text{y}=1,\text{z}=0}$ and $\Pr_{\text{train}}(\text{z} = 1) = w_{\text{y}=1,\text{z}=1} + w_{\text{y}=0,\text{z}=1}$.

As the above optimization is a non-convex quadratically constrained quadratic problem (non-convex QCQP), we now apply the semidefinite relaxation (SDP relaxation) (Park & Boyd, 2017). We first rewrite the above optimization using matrices:

$$\min_{x} \ x^T P_0 x + q_0^T x$$
$$\text{s.t. } x^T P_\alpha x \geq 0, \ x^T P_\beta x \leq 0,$$
$$|q_2^T x - \Pr_{\text{train}}(y = 1)| \leq \gamma_{\text{y}},$$
$$|q_3^T x - \Pr_{\text{train}}(z = 1)| \leq \gamma_{\text{z}}$$
$$q_4^T x = 1, \ 0 \leq x_i \leq 1 \ \forall i$$

where $x = [x_1 \ \ x_2 \ \ x_3 \ \ x_4]^T = [w'_{1,1} \ \ w'_{1,0} \ \ w'_{0,1} \ \ w'_{0,0}]^T$, $q_0 = -2[w_{1,1} \ \ w_{1,0} \ \ w_{0,1} \ \ w_{0,0}]^T$, $q_2 = [1 \ \ 0 \ \ 1 \ \ 0]^T$, $q_3 = [1 \ \ 1 \ \ 0 \ \ 0]^T$, $q_4 = [1 \ \ 1 \ \ 1 \ \ 1]^T$,
$P_0 = \text{diag}(\mathbf{1})$, $P_\alpha = \begin{bmatrix} 0 & -\alpha/2 & 0 & (1-\alpha)/2 \\ -\alpha/2 & 0 & (-1-\alpha)/2 & 0 \\ 0 & (-1-\alpha)/2 & 0 & -\alpha/2 \\ (1-\alpha)/2 & 0 & -\alpha/2 & 0 \end{bmatrix}$, and $P_\beta = \begin{bmatrix} 0 & -\beta/2 & 0 & (1-\beta)/2 \\ -\beta/2 & 0 & (-1-\beta)/2 & 0 \\ 0 & (-1-\beta)/2 & 0 & -\beta/2 \\ (1-\beta)/2 & 0 & -\beta/2 & 0 \end{bmatrix}$.

As $x^T P x = \mathbf{Tr}(P(xx^T))$, we can get the following optimization:

$$\min_{X,x} \ \mathbf{Tr}(XP_0) + q_0^T x$$
$$\text{s.t. } \mathbf{Tr}(XP_\alpha) \geq 0, \ \mathbf{Tr}(XP_\beta) \leq 0$$
$$|q_2^T x - \Pr_{\text{train}}(y = 1)| \leq \gamma_{\text{y}},$$
$$|q_3^T x - \Pr_{\text{train}}(z = 1)| \leq \gamma_{\text{z}}$$
$$q_4^T x = 1, \ 0 \leq x_i \leq 1 \ \forall i,$$
$$X = xx^T$$

where $\mathbf{Tr}(\cdot)$ is the trace and $X = xx^T$.

In the above optimization, the last constraint is non-convex. Thus, we relax the non-convex constraint $X = xx^T$ into $X - xx^T \succeq 0$, which is convex, and then use a Schur complement (Zhang, 2006) to get the final SDP form:

$$\min_{X,x} \ \mathbf{Tr}(XP_0) + q_0^T x$$
$$\text{s.t. } \mathbf{Tr}(XP_\alpha) \geq 0, \ \mathbf{Tr}(XP_\beta) \leq 0$$
$$|q_2^T x - \Pr_{\text{train}}(y = 1)| \leq \gamma_{\text{y}},$$
$$|q_3^T x - \Pr_{\text{train}}(z = 1)| \leq \gamma_{\text{z}}$$
$$q_4^T x = 1, \ 0 \leq x_i \leq 1 \ \forall i,$$
$$\begin{bmatrix} X & x \\ x^T & 1 \end{bmatrix} \succeq 0.$$

When we solve the above SDP relaxation problem using convex optimization solvers, we set the 5×5-matrix A as the variable so as to indicate $\begin{bmatrix} X & x \\ x^T & 1 \end{bmatrix}$. Then, we get the solution $x = [x_1 \ \ x_2 \ \ x_3 \ \ x_4]^T$ by taking the first four elements of the last vector in the resulting matrix A.

## A.8 OPTIONAL STEP FOR MINIMIZING OVERALL DISTRIBUTION CHANGE

Continuing from Sec. 4.2, we explain the details on the optional step in Algorithm 1. We can use the optional step (`MinDistChange`) to find non-uniform data sample weights within each (y, z)-class that minimizes the overall distribution change in terms of the Wasserstein distance (Givens & Shortt,

1984), rather than using identical weights. In the optional step, we first divide the examples in each (y=$y$, z=$z$)-class into two sets {y=$y$, z=$z$, t=0} and {y=$y$, z=$z$, t=1} with equal numbers, where t is a specific feature that can be a criterion for dividing the examples. For example, we can find a median of a non-sensitive attribute $x_1$ and set t = 0 for an example if the $x_1$ value of the example is lower than the median of $x_1$. We set t = 1 otherwise. We then make a candidate set of partial weights $w'_{y=y,z=z,t=t}$ on each (y=$y$, z=$z$, t=$t$)-class, where $\sum w'_{y=y,z=z,t=t} = w'_{y=y,z=z}$. Note that we can extend t beyond the binary setting to find more detailed partial weights. For each partial weight candidate, we calculate the candidate sample-wise weights ($\mathbf{d}_{tmp}$) to ensure that the sample weight sum in each (y=$y$, z=$z$, t=$t$)-class is $w_{y=y,z=z,t=t} \cdot n$, where $n$ is the total number of samples in the original training data. Then, we draw new data $\tilde{D}$ from the original training data $D$ via weighted sampling according to $\mathbf{d}_{tmp}$. Then, we calculate the Wasserstein distance between $\tilde{D}$ and $D$ via an optimal transport technique (Peyré & Cuturi, 2019). As a result, the algorithm returns the final sample-wise weights ($\mathbf{d}_{min}$) that result in the closest distribution from the original data.

---

**Algorithm 2:** MinDistChange

---

**Input:** train data $D$, original ratio $w_{y,z}$, new ratio $w'_{y,z}$
In each (y=$y$, z=$z$)-class, divide the examples into two sets {y=$y$, z=$z$, t=0} and {y=$y$, z=$z$, t=1} with equal numbers
partials = $[0, 1/m, 2/m, ..., 1]$
$\mathbf{W}_{y=y,z=z} \leftarrow []$, $\forall (y,z) \in \mathbb{Y} \times \mathbb{Z}$
**for** *each (y=$y$, z=$z$)-class* **do**
    **for** p in partials **do**
        $w'_{y=y,z=z,t=0} = w'_{y=y,z=z} \cdot p$
        $w'_{y=y,z=z,t=1} = w'_{y=y,z=z} \cdot (1-p)$
        Append $[w'_{y=y,z=z,t=0}, w'_{y=y,z=z,t=1}]$ to $\mathbf{W}_{y=y,z=z}$
candidates $\leftarrow \mathbf{W}_{y=1,z=1} \times \mathbf{W}_{y=1,z=0} \times \mathbf{W}_{y=0,z=1} \times \mathbf{W}_{y=0,z=0}$
minD $\leftarrow$ an initial large value
**for** partial-weight in candidates **do**
    $d_j \leftarrow w'_{y=y,z=z,t=t}/(w_{y=y,z=z} \cdot 0.5), \forall j \in \mathbb{I}_{(y,z,t)}, \forall (y,z,t) \in \mathbb{Y} \times \mathbb{Z} \times \{0,1\}$
    $\mathbf{d}_{tmp} = \{d_i\}_{i=1,...,n}$
    Draw data $\tilde{D}$ from $D$ via weighted sampling w.r.t. $\mathbf{d}_{tmp}$
    wassD $\leftarrow$ Calculate the Wasserstein distance between $\tilde{D}$ and $D$ via optimal transport
    **if** *wassD < minD* **then**
        $\mathbf{d}_{min} = \mathbf{d}_{tmp}$
        minD = wassD
**Output :** $\mathbf{d}_{min}$

---

# B APPENDIX – EXPERIMENTS

## B.1 SIMULATION SETTINGS

Continuing from Sec. 3.1, we explain the details of the simulation. The goal of the simulation is to confirm our theoretical observations by generating various synthetic classifiers that show the full range of possible model performances. To this end, we generate 1000 data samples, where each sample has y and z features. We first set y and z to have a specific correlation. We then generate various synthetic classifiers to have different predicted labels ŷ by varying the probability $\Pr(\hat{y} = 1)$ in each (y, z)-class, regardless of the other input features. Note that we do not train actual models (e.g., logistic regression, SVM). For each synthetic classifier, we measure the accuracy and fairness performances based on y, z, and the classifier's ŷ. We repeat the above procedures while varying y and z to have different correlations. As a result, we get the simulation results in Figure 2 by plotting all the classifier accuracy and fairness performances within each (y, z)-correlation.

## B.2 EMPIRICAL OBSERVATIONS FOR EQUALIZED ODDS

Continuing from Sec. 3.1, we perform an experiment to observe that higher (y, z)-correlation leads to a worse accuracy-fairness tradeoff w.r.t. equalized odds (EO) in practice. Figure 4 shows the accuracy-unfairness performances of the logistic regression model trained using FairBatch (Roh et al., 2021) to improve EO on two synthetic datasets with low and high (y, z)-correlations. As a result, low correlation enables classifiers to attain better accuracy-fairness tradeoffs (i.e., close to the bottom right) w.r.t. EO. This experiment shows that the (y, z)-correlation indeed affects the accuracy-fairness tradeoff w.r.t. EO in practice.

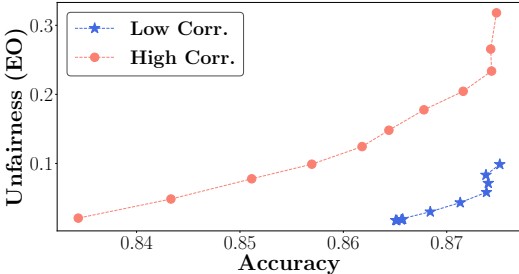

Figure 4: Accuracy-unfairness performances of fair training with FairBatch (Roh et al., 2021) on two synthetic data with different (y, z)-correlations. We measure fairness w.r.t. equalized odds (EO).

## B.3 OTHER EXPERIMENTAL SETTINGS

Continuing from Sec. 5, we provide more details on the experimental settings. The batch sizes of the synthetic, COMPAS, and AdultCensus datasets are 100, 200, and 2000, respectively. For the synthetic dataset, we use 2000 samples for the training dataset and 1000 samples for the test dataset. For the real datasets, we split the entire data into 4:1 for the training and test datasets. We set the learning rate to 0.0005. Our experiments are performed using PyTorch on a Linux server with Intel Xeon Silver 4210R CPUs and NVIDIA Quadro RTX 8000 GPUs.

We generate the synthetic training dataset with 2,000 samples and consists of two non-sensitive attributes $(x_1, x_2)$, one sensitive attribute z, and one label attribute y. Each sample $(x_1, x_2, y)$ is drawn from the following Gaussian distributions: $(x_1, x_2)|y = 0 \sim \mathcal{N}([-2; -2], [10, 1; 1, 3])$ and $(x_1, x_2)|y = 1 \sim \mathcal{N}([2; 2], [5, 1; 1, 5])$. For the sensitive attribute z, we generate a biased distribution: $\Pr(z = 1) = \Pr((x_1', x_2')|y = 1)/[\Pr((x_1', x_2')|y = 0) + \Pr((x_1', x_2')|y = 1)]$ where $(x_1', x_2') = (x_1 \cos(\pi/4) - x_2 \sin(\pi/4), x_1 \sin(\pi/4) + x_2 \cos(\pi/4))$.

When we construct the synthetic test dataset by modifying the z values while fixing the x and y distributions in the original test data, we change the $k$ value in $(x_1', x_2') = (x_1 \cos(\pi/k) - x_2 \sin(\pi/k), x_1 \sin(\pi/k) + x_2 \cos(\pi/k))$.

For all in-processing algorithms, we start from a candidate set and use cross-validation on the (pre-processed) training data to choose the hyperparameters that result in the best fairness while having an accuracy that best aligns with other results.

To support multiple fairness metrics, we naturally extend each in-processing approach by combining the fairness constraints for different metrics via a tuning knob that adjusts the importance between each metric. Here, the fairness constraints are implemented differently in each algorithm. Fairness Constraints (FC) (Zafar et al., 2017a) adds each unfairness penalty term to the loss function. Thus, we extend FC by adding multiple penalty terms and adjust the importance between each penalty term by a tuning knob. Adversarial Debiasing (AD) (Zhang et al., 2018) utilizes a discriminator of each fairness metric for the adversarial training. Thus, we extend AD by adding multiple fairness discriminators and adjust the importance between each discriminator by adding a tuning knob in the classifier's loss function. FairBatch (FB) (Roh et al., 2021) solves a bilevel optimization that has an objective for minimizing a fairness disparity according to each fairness metric. Thus, we extend FB by minimizing the maximum of fairness disparities (e.g., $\max(DPdisp., EOdisp.)$) and adjust the importance between each disparity using a tuning knob.

## B.4 Accuracy and Fairness – AdultCensus

Continuing from Sec. 5.1, we show the results on the AdultCensus dataset. Table 3 shows the accuracy and fairness performances of the algorithms on the AdultCensus test dataset w.r.t. a single metric (DP) and multiple metrics (DP & EO). Other setting are identical to Table 1. We observe consistent results where our framework improves accuracy and fairness of the in-processing-only baselines and also shows better fairness than the two-step baselines using RW.

Table 3: Performances on the AdultCensus test dataset. Other experimental settings are identical to those in Table 1.

| Method | Single (DP) Acc. | Single (DP) Unfair. | Multiple (DP & EO) Acc. | Multiple (DP & EO) Unfair. |
|---|---|---|---|---|
| LR | $.824 \pm .001$ | $.074 \pm .002$ | $.824 \pm .001$ | $.074 \pm .002$ |
| FC | $.805 \pm .013$ | $.021 \pm .004$ | $.807 \pm .013$ | $.078 \pm .017$ |
| RW+FC | $.810 \pm .004$ | $.020 \pm .006$ | $.810 \pm .008$ | $.048 \pm .013$ |
| **Ours**+FC | $.820 \pm .003$ | $\mathbf{.005 \pm .002}$ | $.805 \pm .014$ | $\mathbf{.033 \pm .013}$ |
| AD | $.777 \pm .022$ | $\mathbf{.007 \pm .004}$ | $.806 \pm .011$ | $.052 \pm .004$ |
| RW+AD | $.808 \pm .010$ | $.025 \pm .007$ | $.800 \pm .015$ | $.035 \pm .003$ |
| **Ours**+AD | $.803 \pm .009$ | $\mathbf{.007 \pm .003}$ | $.805 \pm .009$ | $\mathbf{.033 \pm .003}$ |
| FB | $.825 \pm .002$ | $.017 \pm .001$ | $.826 \pm .001$ | $.049 \pm .001$ |
| RW+FB | $.818 \pm .008$ | $.020 \pm .005$ | $.817 \pm .008$ | $.071 \pm .010$ |
| **Ours**+FB | $.824 \pm .001$ | $\mathbf{.008 \pm .003}$ | $.826 \pm .003$ | $\mathbf{.037 \pm .012}$ |

## B.5 How the Pre-processed Data Aligns with the True Test Data

Continuing from Sec. 5.1, we also show that the pre-processed data distribution ($D_{\text{pre}}$) by our algorithm is more aligned with the true test distribution ($D_{\text{test}}$) compared to the original training distribution ($D_{\text{train}}$), in terms of $(y, z)$-correlation and Wasserstein distance. We calculate the second-order Wasserstein distance via an optimal transport technique (Peyré & Cuturi, 2019) on the synthetic data. We experiment on three degrees on shifts and make two observations in Table 4: 1) the correlations of $D_{\text{pre}}$ and $D_{\text{test}}$ are indeed more similar relative to $D_{\text{train}}$ and 2) the Wasserstein distance between $D_{\text{pre}}$ and $D_{\text{test}}$ is lower than between $D_{\text{train}}$ and $D_{\text{test}}$. Both observations confirm that the reweighed data aligns well with the test data.

As our method improves the alignment between $D_{\text{pre}}$ and $D_{\text{test}}$, the in-processing algorithms trained on $D_{\text{pre}}$ (i.e., ours + in-processing algorithms) show high accuracy and fairness performances on $D_{\text{test}}$, as shown in Sec. 5.

Table 4: Alignment between data distributions. We use the synthetic data.

| | Level of correlation shift in $c_{\text{test}}$ Severe | Level of correlation shift in $c_{\text{test}}$ Normal | Level of correlation shift in $c_{\text{test}}$ No shift |
|---|---|---|---|
| $c_{\text{train}}$ (correlation in $D_{\text{train}}$) | 0.3591 | 0.3591 | 0.3591 |
| $c_{\text{pre}}$ (correlation in $D_{\text{pre}}$) | 0.0359 | 0.1796 | 0.3590 |
| $c_{\text{test}}$ (correlation in $D_{\text{test}}$) | 0.0360 | 0.1800 | 0.3591 |
| Wass. dist. between $D_{\text{train}}$ and $D_{\text{test}}$ | 0.7966 | 0.6807 | 0.4399 |
| Wass. dist. between $D_{\text{train}}$ and $D_{\text{test}}$ | 0.6257 | 0.6022 | 0.4399 |

## B.6 Accuracy and Fairness – Other Test Data Construction: Synthetic Data

Continuing from Sec. 5.1, we compare the algorithm performances when using a different method to construct the test dataset of the synthetic-data experiment. Table 5 shows the accuracy and fairness performances on the synthetic dataset when constructing the test dataset by modifying z directly.

Table 5: Performances on the synthetic test dataset. The test dataset is constructed via modifying the z values of the original distribution. Other settings are identical to those in Table 1.

| | Single (DP) | | Multiple (DP & EO) | |
|---|---|---|---|---|
| Method | Acc. | Unfair. | Acc. | Unfair. |
| LR | $.871 \pm .000$ | $.138 \pm .000$ | $.871 \pm .000$ | $.138 \pm .000$ |
| FC | $.805 \pm .006$ | $.040 \pm .005$ | $.830 \pm .001$ | $.119 \pm .003$ |
| RW+FC | $.847 \pm .005$ | $.047 \pm .007$ | $.852 \pm .005$ | $.053 \pm .004$ |
| **Ours**+FC | $.831 \pm .003$ | $\mathbf{.005 \pm .004}$ | $.854 \pm .004$ | $\mathbf{.052 \pm .002}$ |
| AD | $.792 \pm .012$ | $.048 \pm .009$ | $.815 \pm .008$ | $.133 \pm .006$ |
| RW+AD | $.836 \pm .010$ | $.042 \pm .008$ | $.849 \pm .004$ | $\mathbf{.054 \pm .004}$ |
| **Ours**+AD | $.820 \pm .006$ | $\mathbf{.005 \pm .003}$ | $.847 \pm .002$ | $.057 \pm .007$ |
| FB | $.804 \pm .001$ | $.051 \pm .002$ | $.831 \pm .001$ | $.128 \pm .002$ |
| RW+FB | $.844 \pm .004$ | $.036 \pm .003$ | $.853 \pm .005$ | $.057 \pm .011$ |
| **Ours**+FB | $.824 \pm .002$ | $\mathbf{.015 \pm .002}$ | $.854 \pm .003$ | $\mathbf{.054 \pm .003}$ |

We again generate the synthetic dataset using a method similar to Zafar et al. (2017a). The synthetic dataset consists of two non-sensitive attributes $(x_1, x_2)$, one sensitive attribute z, and one label attribute y. Each sample $(x_1, x_2, y)$ is drawn from the following Gaussian distributions: $(x_1, x_2)|y = 0 \sim \mathcal{N}([-2; -2], [10, 1; 1, 3])$ and $(x_1, x_2)|y = 1 \sim \mathcal{N}([2; 2], [5, 1; 1, 5])$. For the sensitive attribute z, we generate a biased distribution: $\Pr(z = 1) = \Pr((x_1', x_2')|y = 1)/[\Pr((x_1', x_2')|y = 0) + \Pr((x_1', x_2')|y = 1)]$ where $(x_1', x_2') = (x_1 \cos(\pi/k) - x_2 \sin(\pi/k), x_1 \sin(\pi/k) + x_2 \cos(\pi/k))$. For the training dataset, we set $k$ to 4. We then change the $k$ value to generate the test dataset with 50% of the training data correlation.

As a result, we observe consistent results where our framework improves the accuracy and fairness performances of the in-processing-only baselines and generally shows better fairness than the two-step baselines using RW.

### B.7 ACCURACY AND FAIRNESS – OTHER TEST DATA CONSTRUCTION: REAL DATA

Continuing from Sec. 5.1, we compare the algorithm performances when using a different method to construct the test dataset of the real-data experiment. Table 6 shows the accuracy and fairness performances when using the two income datasets collected in the 1990s (Kohavi, 1996) (i.e., AdultCensus dataset used in Sec. B.4) and 2010s (Ding et al., 2021) for training and testing, respectively. Here, the 1990s and 2010s datasets have the different $(y, z)$-correlation values 0.214 and 0.150, respectively. In addition, the input feature x distribution also slightly shifted, as shown in Figure 5. As a result, we observe consistent results in Table 6 where our framework improves the accuracy and fairness performances of the in-processing-only baselines.

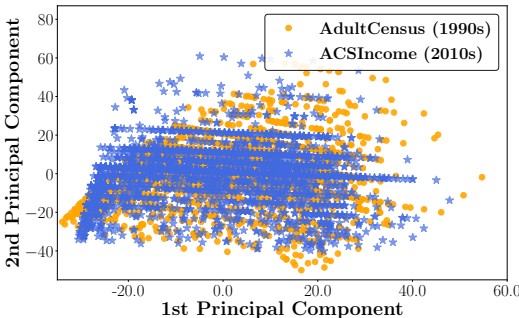

Figure 5: Principal component analysis (PCA) results on the input feature x of the two US income datasets: AdultCensus (Kohavi, 1996) and ACSIncome (Ding et al., 2021). Here, the distributions of the principal components on input feature x are different in the two datasets.

Table 6: Model performances on the ACSIncome test data (Ding et al., 2021) collected in the 2010s. The models are trained on the AdultCensus data (Kohavi, 1996) collected in the 1990s. Both datasets have labels that indicate each person's annual income.

|  | Single (DP) | |
| --- | --- | --- |
| Method | Acc. | Unfair. |
| FC | $.646 \pm .013$ | $.104 \pm .015$ |
| **Ours**+FC | $.652 \pm .015$ | $.090 \pm .017$ |
| AD | $.637 \pm .031$ | $.137 \pm .037$ |
| **Ours**+AD | $.644 \pm .024$ | $.060 \pm .039$ |
| FB | $.643 \pm .007$ | $.128 \pm .021$ |
| **Ours**+FB | $.670 \pm .004$ | $.098 \pm .002$ |
| *FB on test dist. (upper bound)* | $.704 \pm .007$ | $.044 \pm .035$ |

## B.8 ACCURACY AND FAIRNESS – TRADEOFF CURVES

Continuing from Sec. 5.1, we provide the accuracy-fairness disparity (DP) trade-off curves of FairBatch (FB) and Ours+FB on the the synthetic data in Figure 6. Our framework shows a better accuracy-fairness tradeoff compared to the in-processing approach (FB). FB shows an "inversed" curve, as it is trained with wrong biases in mind.

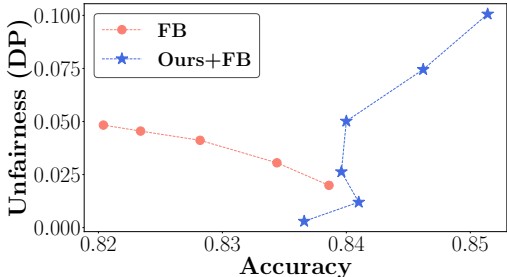

Figure 6: Accuracy-fairness disparity trade-off curves of FB and Ours+FB on the synthetic data.

## B.9 USING THE OPTIONAL STEP

 we show the results of when using the optional step using `MinDistChange` in Algorithm 1. This step finds possibly-different data sample weights within each $(y, z)$-class to minimize the overall distribution change between the training and pre-processed data. Figure 7 shows the Wasserstein distances between the original training data and pre-processed data when our algorithm uses either the basic or optional step, and the correlation constant $c$ ranges from 10% to 70% of the training data's correlation. As a result, the optional step generally reduces the Wasserstein distance between the training and pre-processed data distributions, especially for smaller $c$ values. In addition, Table 7 shows the performances on the synthetic test dataset when our algorithm uses either the basic step (Ours) or the optional step (Ours+Optional). As a result, when using the new data from the optional step as an input of the in-processing approaches, the final classifier's accuracy and fairness are not sacrificed much compared to using data from the basic step, while minimally changing the data distribution.

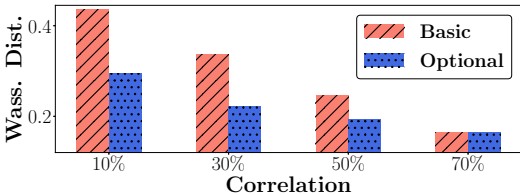

Figure 7: Wasserstein distances between the original training data and pre-processed data when our algorithm uses either the basic step or the optional step, while varying the target correlation to have 10% to 70% correlation of the training data.

Table 7: Performances on the synthetic test dataset. Other settings are identical to those in Table 1.

| Method | Single (DP) | |
| --- | --- | --- |
| | Acc. | Unfair. |
| **Ours**+FC | $.849 \pm .002$ | $.034 \pm .004$ |
| **Ours+Optional**+FC | $.830 \pm .003$ | $.027 \pm .005$ |
| **Ours**+AD | $.814 \pm .011$ | $.017 \pm .006$ |
| **Ours+Optional**+AD | $.808 \pm .007$ | $.014 \pm .006$ |
| **Ours**+FB | $.836 \pm .001$ | $.003 \pm .001$ |
| **Ours+Optional**+FB | $.842 \pm .002$ | $.010 \pm .004$ |

## B.10 SETTING THE CORRELATION RANGE TO $c_{\text{TEST}} \pm x\%$

Continuing from Sec. 5.2, we evaluate our approach when $[\alpha, \beta] = [c_{\text{test}} - x\%, c_{\text{test}} + x\%]$. Figure 8 shows the accuracy and fairness performances of FB and Ours+FB, while varying the correlation constant $c$ of the test data. Here, we set the *specified* correlation range in our algorithm to $[\alpha, \beta] = [c_{\text{test}} - x\%, c_{\text{test}} + x\%]$, where $x \in \{10, 50, 100\}$. When $x = 10\%$, we observe similar trends as in Figure 3a, where our framework improves the accuracy and fairness performances of the in-processing-only baselines. When $x = 100\%$ (i.e., the worst-case setting of $[\alpha, \beta]$), the accuracy and fairness performances of our framework converge to the in-processing-only baselines. When $x = 50\%$, the performances of our framework are in between those of when $x = 10\%$ and $x = 100\%$. There are two takeaways: 1) our framework successfully boosts the in-processing-only baseline performances when the $[\alpha, \beta]$ range is reasonable, and 2) even if we do not have any information about the correlation shift, our framework performs at least as well as the in-processing-only baselines.

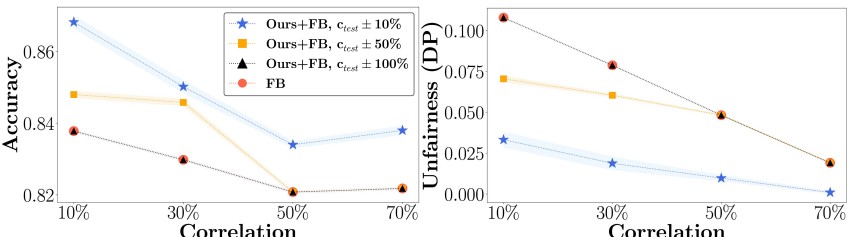

Figure 8: Performances of FB and Ours+FB on the synthetic data while varying the correlation of the test data $c_{\text{test}}$ to have 10% to 70% correlation of the training data. We set the *specified* correlation range in our algorithm to $[\alpha, \beta] = [c_{\text{test}} - x\%, c_{\text{test}} + x\%]$, where $x \in \{10, 50, 100\}$.

## B.11 VARYING THE CORRELATION OF THE TEST DATA: MORE BIASES

Continuing from Sec. 5.2, we vary the correlation of the test data to have more biases than the training data. Figure 9 shows the accuracy and fairness performances of FB and Ours+FB, while varying the correlation constant $c$ of the test data up to 150%. When the correlation increases more than 100% of that of the training data, the in-processing-only baseline (FB) cannot improve fairness because the training data does not capture the bias level in the test data. Hence, the in-processing-only baseline cannot be used in applications that require high fairness. On the other hand, our pre-processing enables the in-processing approach to achieve the high fairness that it may need, with some accuracy degradation.

## B.12 EMPIRICAL COMPARISON WITH GIGUERE ET AL. (2022)

Continuing from Sec. 6, we add a new baseline called Shifty(Giguere et al., 2022), which focuses on the distribution shift most relevant to ours. Shifty first trains candidate models on the training data and selects only the models showing high fairness in the shifted deployment data. To give a favorable condition to Shifty, we assume that Shifty knows the exact test distribution. Table 8 shows the accuracy and fairness performances of the in-processing-only baseline FairBatch, Shifty, and our framework w.r.t. a single metric (DP) and multiple metrics (DP & EO) in the synthetic and COMPAS datasets used in Table 1. As a result, both ours and Shifty improve the fairness of the

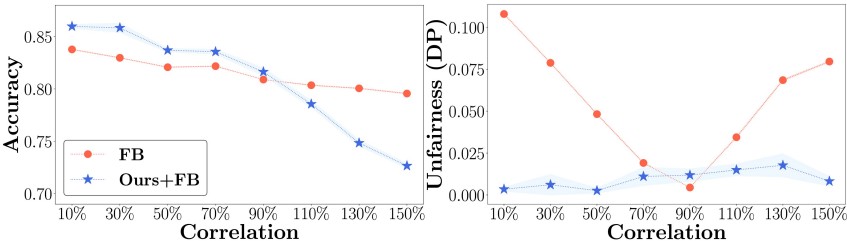

Figure 9: Performances of FB and Ours+FB on the synthetic data while varying the correlation of the test data to have 10% to 150% correlation of the training data.

Table 8: Performances on the synthetic test dataset. We compare in-processing-only baseline FairBatch (Roh et al., 2021), Shifty (Giguere et al., 2022), and our framework. Other settings are identical to those in Table 1.

|  | Synthetic | | | | COMPAS | | | |
|---|---|---|---|---|---|---|---|---|
|  | Single (DP) | | Multiple (DP & EO) | | Single (DP) | | Multiple (DP & EO) | |
| Method | Acc. | Unfair. | Acc. | Unfair. | Acc. | Unfair. | Acc. | Unfair. |
| FB (Roh et al., 2021) | .821 ± .000 | .048 ± .000 | .849 ± .001 | .091 ± .005 | .647 ± .001 | .038 ± .013 | .650 ± .002 | .187 ± .019 |
| FB + Shifty | **.838 ± .001** | .024 ± .008 | .844 ± .003 | .063 ± .008 | .647 ± .001 | .029 ± .001 | .649 ± .002 | .162 ± .025 |
| **Ours+FB** | .836 ± .001 | **.003 ± .001** | **.852 ± .004** | **.058 ± .001** | **.648 ± .004** | **.027 ± .001** | **.657 ± .004** | **.130 ± .014** |

in-processing-only baseline, but ours shows better fairness than Shifty while achieving similar or higher accuracy. The reason is that Shifty is selecting the final model among the candidates that were already trained on the original training data, whereas ours trains a new model on the improved (pre-processed) data.

### B.13 SUPPORTING NOISY GROUP ATTRIBUTES

Continuing from Sec. 6, we evaluate the performance of our method on increasingly noisy data and observe that our method can potentially be extended to the noisy group scenario.

We perform a new experiment, where 10–50% of the group information in the training data is randomly flipped. Table 9 shows the accuracy and fairness performances of the in-processing-only baseline FairBatch (FB) and our framework (Ours+FB) w.r.t. demographic parity (DP). Here, FB cannot achieve high fairness performance (i.e., low DP disp.), as the group distribution in the training data is different from that in the test data. In comparison, our framework enables FB to achieve relatively high fairness at the expense of some accuracy degradation.

We note that these results are preliminary and only show that our method has some potential to support noisy group attributes, and we believe there is plenty of room for improvement. In particular, we believe our method can be further extended with other robust training methods like (Shen & Sanghavi, 2019).

Table 9: Performances on the synthetic test dataset when the group information in the training data is randomly flipped. We train the algorithms w.r.t. demographic parity (DP).

|  | 10% flipping | | 30% flipping | | 50% flipping | |
|---|---|---|---|---|---|---|
| Method | Acc. | Unfair. (DP) | Acc. | Unfair. (DP) | Acc. | Unfair. (DP) |
| FB | 0.818 | 0.087 | 0.864 | 0.165 | 0.883 | 0.250 |
| **Ours** + FB | 0.808 | 0.069 | 0.823 | 0.101 | 0.838 | 0.142 |

### B.14 SUPPORTING POISONING ATTACK SCENARIO

Continuing from Sec. 6, we evaluate the performance of our method on poisoned data and observe that our method can potentially be extended to the poisoning attack scenario.

We perform a new experiment using one of the poisoning attack methods for fair training (Roh et al., 2020), where 10% of the training labels of a specific group are flipped so as to maximize the accuracy

degradation. Table 10 shows the accuracy and fairness performances of the in-processing-only baseline FairBatch (FB) and our framework w.r.t. demographic parity (DP) on the poisoned data; we also report FB on the clean data, which can be considered as the upper-bound performance. Here, when achieving similar fairness, our framework improves the accuracy of FB on the poisoned data. We suspect that, while our pre-processing reduces the correlation shift between training and deployment data, the effect of the poisoning attack is also mitigated.

We note that these results are preliminary and only show that our method has some potential to support the poisoning attack scenario, and we believe there is plenty of room for improvement.

Table 10: Performances on the synthetic test dataset when 10% of the training labels are poisoned (Roh et al., 2020). We train the algorithms w.r.t. demographic parity (DP).

| Method | Acc. | Unfair. (DP) |
|---|---|---|
| FB | 0.699 | 0.026 |
| **Ours** + FB | 0.721 | 0.024 |
| *FB on the clean data (upper bound)* | 0.773 | 0.023 |

## C   APPENDIX – MORE RELATED WORK

### C.1   TRADITIONAL MODEL FAIRNESS

As model fairness becomes essential for Trustworthy AI, various fairness definitions have been proposed to reflect legal and social requirements (Narayanan, 2018). Among the definitions, we focus on group fairness, which aims to not discriminate specific groups. There are three prominent group fairness metrics: demographic parity (Feldman et al., 2015), equalized odds (Hardt et al., 2016), and predictive parity (Berk et al., 2021). To support the fairness metrics, various fairness techniques have been proposed, which can be categorized into three prominent approaches: (1) pre-processing approaches (Kamiran & Calders, 2011; Zemel et al., 2013; Feldman et al., 2015; du Pin Calmon et al., 2017; Choi et al., 2020; Jiang & Nachum, 2020), which debias, reweight, or generate training data, (2) in-processing approaches (Zafar et al., 2017a;b; Agarwal et al., 2018; Zhang et al., 2018; Cotter et al., 2019; Roh et al., 2020; 2021), which modify model training itself for fairness, and (3) post-processing approaches (Kamiran et al., 2012; Hardt et al., 2016; Pleiss et al., 2017; Chzhen et al., 2019), which manipulate only the model outputs without changing the training inside. Among the three categories, in-processing approaches are widely used for unfairness mitigation, but most of them assume that the training and deployment data distributions are the same.

Another line of research is to support multiple fairness metrics (Thomas et al., 2019; Zhao et al., 2020). Thomas et al. (2019) proposes a fairness testing framework that can support multiple metrics. Zhao et al. (2020) shows that EO can be achieved while preserving the original DP. In comparison, we analyze when a model can achieve both $\varepsilon$-DP and $\varepsilon$-EO.

Beyond group fairness, there are other noteworthy fairness definitions including individual fairness (Dwork et al., 2012b), which aims to give similar predictions to similar individuals, and causality-based fairness (Kilbertus et al., 2017; Kusner et al., 2017; Zhang & Bareinboim, 2018), which aims to improve fairness by understanding the causal relationship between attributes. Extending our analysis and framework to these definitions is an interesting future work.

### C.2   FAIRNESS UNDER DATA DISTRIBUTION SHIFTS

Continuing from Sec. 6, we further compare our work with the previous studies on data distribution shifts. The following paragraphs contain detailed discussions of the two categories of distribution shifts: general distribution shifts and fairness-specific shifts.

Among the general distribution shifts (i.e., covariate, label, and concept shifts in Table 2), the concept shift is the most relevant definition to the correlation shift. As shown in Table 2 of Sec. 6, the correlation and concept shifts focus on $\Pr(z|y)$ and $\Pr(y|x)$, respectively. As the concept shifts consider the distribution changes of the label (y) and input feature (x), this type of shift can implicitly

describe the correlation shifts when the input feature contains the group attribute (z). However, a unique characteristic of the correlation shifts is to explicitly capture the bias changes between y and z, where z is especially relevant to fair training. Thus, the notion of correlation shifts enables us to analyze the behavior of fair training when the bias changes.

In addition, the key difference between the correlation shift and the other fairness-specific shifts (subpopulation and demographic shifts) is that the other shifts are defined under specific assumptions on the data distribution, which are not required in our correlation shift definition. Here are the assumptions in the subpopulation and demographic shifts:

- The subpopulation shift (Maity et al., 2021) assumes that the loss expectations w.r.t. input feature X and label Y of training and deployment distributions are the same (i.e., $E_{\text{train}}[(h(\text{x}), \text{y})|\text{z} = z] = E_{\text{test}}[(h(\text{x}), \text{y})|\text{z} = z]$, where $E$ indicates the expectation). Thus, as discussed in the related work section, a major example of subpopulation shifts is when a specific group has fewer positively-labeled examples during training time compared to deployment time, while the distributions of the x and y attributes remain the same.

- Similarly, the demographic shift (Giguere et al., 2022) assumes that the joint probabilities of x and y on the training and deployment distributions are identical (i.e., $\text{Pr}_{\text{train}}(\text{x} = x, \text{y} = y|\text{z} = z) = \text{Pr}_{\text{test}}(\text{x} = x, \text{y} = y|\text{z} = z)$). This assumption is similar to the assumption of the subpopulation shift, but the difference is whether the loss values of the model are explicitly considered or not.

The above assumptions are used in the theoretical analyses of the previous works (Maity et al., 2021; Giguere et al., 2022). In comparison, our theoretical analyses in Sec. 3 of using correlation shifts are not limited by any additional assumptions, and we show that the achievable performances of fair training are determined by the (y, z)-correlation.

### C.3 CONNECTION TO CAUSALITY-BASED FAIRNESS

Continuing from Sec. 6, we discuss how our work can be connected with causality-based fairness. To this end, we give a concrete example to show the correlation shifts when the sensitive group attribute itself is a confounder, one of the important roles of attributes in causality-based fairness.

We consider a car insurance example, where the goal is to predict future accident rates from the past driving record: suppose that input feature x is the past driving record, label y is the future accident rate, and sensitive attribute z is the race.

Then, the structural causality graph between them would look like: z → (x, y) and x → y, i.e., z being a confounder (Figure 10). The arrows from z to x and y could be due to (a) different socialization and driving behavior formation process that partially depends on racial identity and (b) another unobserved mediator such as where they live in, etc. In this example where z is a confounder, it is clear that z and y are correlated.

We expect this confounding effect to change over time. This will make the sensitive attribute z either a stronger or weaker confounding factor, which impacts the correlation between y and z.

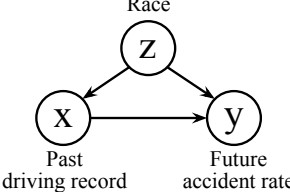

Figure 10: A causality graph for the car insurance example in Sec. C.3.

