# OpenReview forum: "The World is Changing: Improving Fair Training under Correlation Shifts"
_ICLR.cc/2023/Conference — Submitted to ICLR 2023_

### Official Review · Reviewer_c8Rf · 2022-10-22

**Confidence:** 4
**Correctness:** 4
**Technical Novelty And Significance:** 3
**Empirical Novelty And Significance:** 4
**Recommendation:** 8

**Clarity, Quality, Novelty And Reproducibility:**

The proposed idea is sound, the paper is well written. The proposed method, in my opinion, is reproducible.

**Strength And Weaknesses:**

Advantages:
- This article is well-written in general and addresses a timely, and seemingly overlooked, issue in fairness.
- Solid theoretical and simulation results.
- Strong mitigation performance.

Drawbacks:
- One minor concern is that the authors only consider scenarios in which we know the deployment data distributions. Will the proposed solution be useful despite our lack of knowledge about deployment data distributions?
- Furthermore, it would be great if the authors could demonstrate the results using the fairness and accuracy trade-off curve rather than a single point on the curve. This could help to better demonstrate the effectiveness of the proposed method.


**Summary Of The Paper:**

While many techniques for model fairness have been proposed, the majority of them assume that the distributions of training and deployment data are identical, which is often not the case in practice. In particular, the bias between labels and sensitive groups changes, which may impair the performance of machine learning algorithms. To this end, this paper proposes a novel fair training framework that is divided into two parts: using a pre-processing approach to reflect the shifted correlation and using any existing in-processing algorithms for fair training on top of the improved data. Experimental results have validated the effectiveness of the proposed method under correlation shifts.


**Summary Of The Review:**

This paper works on an timely problem, has a solid theoretical analysis and strong experimental results. I recommend acceptance of this paper.

---

> ### Author Response · Authors · 2022-11-16
> **Response to Reviewer c8Rf**
>
> We really appreciate your positive comments and thoughtful suggestions. We addressed them below and improved our manuscript.
>
> ------------------------------
> **Comment 1: Algorithm behavior when knowledge of deployment distribution is lacking**
>
> We appreciate this comment and would like to clarify that our method can still work when the knowledge of deployment data distribution is lacking, as we have indeed shown in the paper. We considered two related scenarios: specifying the totally incorrect correlations in Figure 3b (i.e., c_specified $\neq$ c_test) and using the ranges of correlation in Figure 6 (i.e., c_specified $\in$ [c_test - x%, c_test + x%]). In both scenarios, our framework successfully boosts the in-processing-only baseline’s performances when the estimation error is respectful (say up to $\pm$ 10%). Also, Figure 6 shows that even if we do not have any information about the shift (i.e., c_specified $\in$ [c_test - 100%, c_test + 100%]), ours performs at least as well as the in-processing-only baseline.
>
> We further clarified these results in our revision (Section 5.3, highlighted in blue).
>
> ------------------------------
> **Comment 2: Fairness and accuracy trade-off curve**
>
> We fully agree that the fairness and accuracy trade-off curves can better demonstrate the effectiveness of the proposed method. We actually have this result using synthetic data in Figure 5, Section B.8. Here our framework shows a better tradeoff compared to the in-processing-only baseline FairBatch, which shows an “inversed” curve, as it is trained with wrong biases in mind.
>
> Per your great comment, we now emphasized this result by clearly connecting it to the main body of the paper (Section 5.1, highlighted in blue).

---

### Official Review · Reviewer_9fri · 2022-10-25

**Confidence:** 4
**Correctness:** 3
**Technical Novelty And Significance:** 3
**Empirical Novelty And Significance:** 3
**Recommendation:** 5

**Clarity, Quality, Novelty And Reproducibility:**

The paper studies a novel and interesting problem, i.e., correlation shift. But the writing needs much improvement, which is sometimes vague and lack of important contents.

**Strength And Weaknesses:**

Strengths:
- Distributional shifts is indeed an important problem in machine learning.
- The proposed technique is able to handle correlation shift in the given settings.

Concerns/Questions:
- When we consider distributional shifts, should we only consider the joint distribution of label y and sensitive attribute z? I am quite surprised that we can study it without any assumption on the distribution of input feature x. The model is essentially modeling $P(y|x, z, \theta)$ (or $P(x,y,z|\theta)$ depending on whether the model is discriminative or not). It seems likely that $P(y,z)$ and the marginal distributions are the same but the underlying input feature's distribution $P(x)$ is different, which might still cause bad performance on either accuracy/fairness.
- Is there any relationship between the proposed correlation shift and the concept shift in many OOD literature?
- I am not sure how to read the simulation results in section 3. Is each blue/red dot correspond to one classifier? How are the so-called synthetic classifier generated? And what are their architectures (logistic regression, SVM, MLP)?
- As mentioned in several sections, there are existing works on other types of distributional shifts. I think some of them should be considered as important baseline methods.
- There should be a dedicated paragraph/section to discuss the related work and how this work differs with them.
- Minor comments: (1) fairness is a broad area which is not only group fairness, so it is better to specify that the work is on group fairness rather than just 'fairness'; (2) before using an abbreviation, its full name should be written out first to avoid confusion (e.g., AI -> artificial intelligence (AI)); (3) please carefully check typos or grammatical errors in the revised version.
==== After rebuttal ====
I think the concern on x's distribution remains, e.g., how should we define the 'no drastic change in the distribution of x' more formally? Would it be too strict to hold in real-world scenarios? Why (or why not)?

**Summary Of The Paper:**

This paper studies the distributional shift problem in fair machine learning. The authors first propose a type of distributional shift called correlation shift, which studies the Pearson’s correlation coefficient between the label and the sensitive attribute. The connection between the correlation shift and the notorious accuracy-fairness tradeoff. Based on that, the authors propose a pre-processing strategy that adjusts the ratio of data to correct the shift. Experimental results demonstrate the efficacy of the proposed technique.

**Summary Of The Review:**

The studied problem is important as it helps with the generalization of fair machine learning methods. However, I have concerns regarding its motivation and settings for analysis.

---

> ### Author Response · Authors · 2022-11-16
> **Response to Reviewer 9fri (Part 1)**
>
> We appreciate your thoughtful comments and useful suggestions. We addressed them below and improved our manuscript.
>
> --------------------------
> **Concerns/Questions 1: Assumption on the distribution of input feature x**
>
> As per your great comment, we clarify how we consider the input feature x: our framework (Section 4) implicitly assumes the distribution of x does not change drastically, but we empirically observe that our framework performs well in the real-world scenario when the x distribution changes (Section 5).
>
> Our framework is designed to work best when the x distribution does not change, but does not strictly require this condition unlike other previous works on fairness-specific shifts [1,2]. Specifically, Algorithm 1 uses random sampling w.r.t. the new weights on each (y, z)-class, so it is desirable that the x distribution conditioned on each (y, z)-class stays the same. Of course, if the x distribution completely changes, we agree that the fairness or accuracy will degrade. We thus added clarifications in the paper that we implicitly assume that the x distribution does not change drastically and focus on the (y, z)-correlation.
>
> In our paper, we empirically show that our framework indeed performs well in real-world x distribution shift scenarios. For example, Table 6 in Section B.7 shows the accuracy and fairness performances when using the two income datasets collected in the 1990s [3] (i.e., AdultCensus dataset) and 2010s [4] (i.e., ACSIncome dataset) for training and testing, respectively. Since the two datasets are collected separately in different periods, both the (y, z)-correlation and the x distribution can be different between these datasets. In Sec. B.7, we indeed observe that the 1990s and 2010s datasets have different (y, z)-correlation values 0.214 and 0.150, respectively, and that the input feature x distribution also shifted when investigating it via principal component analysis (PCA). Nevertheless, we observe that our framework successfully improves the accuracy and fairness performances of the in-processing-only baselines.
>
> We added these points in our revision (Section 4 and Section A.6, highlighted in blue).
>
> [1] Maity et al., “Does enforcing fairness mitigate biases caused by subpopulation shift?”, NeurIPS’21
>
> [2] Giguere et al., “Fairness guarantees under demographic shift”, ICLR’22
>
> [3] Kohavi, “Scaling up the accuracy of naive-bayes classifiers: A decision-tree hybrid”, SIGKDD’96
>
> [4] Ding et al., “Retiring adult: New datasets for fair machine learning”, NeurIPS’21
>
> --------------------------
> **Concerns/Questions 2: Relationship between the correlation shifts and concept shifts**
>
> As per your important question, we further discuss the relationship between the correlation and concept shifts. As in Table 2 of the related work section, the correlation and concept shifts focus on Pr(z|y) and Pr(y|x), respectively. As the concept shifts consider the distribution changes of the label (y) and input feature (x), this type of shift can implicitly describe the correlation shifts when the input feature contains the group attribute (z). However, a unique characteristic of the correlation shifts is to explicitly capture the bias changes between y and z, where z is especially relevant to fair training. Thus, the notion of correlation shifts enables us to analyze the behavior of fair training when the bias changes.
>
> We clarified this point in our revision (Section C.2, highlighted in blue).
>
> --------------------------
> **Concerns/Questions 3: Clarification of the simulation results**
>
> We appreciate your questions and clarify as follows. Each red and blue point indeed represents a single classifier. Instead of training actual models (e.g., logistic regression, SVM), we design synthetic classifiers to have different predicted labels $\hat{y}$ by varying the probability $\Pr(\hat{y}=1)$ in each (y, z)-class, regardless of the other input features. We then measure the accuracy and fairness of each classifier based on its $\hat{y}$ values.
>
> We clarified the simulation setting in our revision (Figure 2 in Section 3 and Section B.1, highlighted in blue).

---

> > ### Author Response · Authors · 2022-11-16
> > **Response to Reviewer 9fri (Part 2)**
> >
> > **Concerns/Questions 4: Additional baseline designed for other distribution shifts**
> >
> > As per your helpful comment, we added a new baseline called Shifty [1], which focuses on the distribution shift most relevant to ours, as discussed in our related work section. Shifty first trains candidate models on the training data and selects only the models showing high fairness in the shifted deployment data. To give a favorable condition to Shifty, we assume that Shifty knows the exact test distribution. Table 8 in our revision shows the accuracy and fairness performances of the in-processing-only baseline FairBatch, Shifty, and our framework w.r.t. a single metric (DP) and multiple metrics (DP & EO) in the synthetic and COMPAS datasets used in Table 1. As a result, both ours and Shifty improve the fairness of the in-processing-only baseline, but ours shows better fairness than Shifty while achieving similar or higher accuracy. The reason is that Shifty is selecting the final model among the candidates that were already trained on the original training data, whereas ours trains a new model on the improved (pre-processed) data.
> >
> > We added this result in our revision (Section B.12, highlighted in blue).
> >
> > [1] Giguere et al., “Fairness guarantees under demographic shift”, ICLR’22
> >
> > ------------------------------
> > **Concerns/Questions 5: Related work**
> >
> > Based on your previous comments together with Reviewer H3mp’s suggestion, we enriched the discussion in related work by further comparing ours with the previous studies on data distribution shifts. The main discussion from our original manuscript is in Section 6 (paragraphs 2–4 and Table 2), and the new discussion is in Section C.2 (highlighted in blue).
> >
> > ------------------------------
> > **Concerns/Questions 6: Minor comments**
> >
> > We do appreciate your suggestions to improve the presentation quality. We made our best efforts to address all your points in our revision. We now clarified that we focus on group fairness, carefully stated full expressions before using abbreviations, and improved the overall writing.

---

> > > ### Comment · Reviewer_9fri · 2022-11-19
> > > **Feedback**
> > >
> > > Thank you for your efforts in addressing my concerns. I think most concerns are addressed. But I think the implicit assumption on x's distribution needs more discussion, e.g., how should we define the 'do not change drastically' more formally? Would it be too rigorous in many real-world cases?
> > >
> > > Other than that, my other concerns are well addressed, and I will raise my score a bit. But given that my major concern still remains, I am still not championing this paper.

---

> > > > ### Author Response · Authors · 2022-11-20
> > > > **To Reviewer 9fri**
> > > >
> > > > We really appreciate your response and suggestions for more discussion on the input feature x distribution. We agree our paper can be strengthened if we formally define the allowed x distribution change or discuss possible limitations in real-world applications. Indeed, although we anticipate that our framework will work well in many real-world scenarios, there may be some scenarios where our framework does not achieve expected performances, e.g., when the input feature ranges of the training and deployment distributions do not overlap at all. As per your important comment, we will discuss such limitations in our revision and suggest future work on how to relax the implicit assumption in our framework.
> > > >
> > > > Thank you again for your valuable comments, and we will do our best to address them in our revision.

---

### Official Review · Reviewer_gS56 · 2022-10-28

**Confidence:** 3
**Clarity, Quality, Novelty And Reproducibility:** The paper is generally well-written a…
**Correctness:** 3
**Technical Novelty And Significance:** 3
**Empirical Novelty And Significance:** 3
**Recommendation:** 6

**Strength And Weaknesses:**

Strengths:
An important problem is addressed here. It is valuable to understand how bias affects fairness properties. The paper is generally well-written.

Weakness:

1. The correlation between the sensitive attribute and the label could be more nuanced depending on the role of the sensitive attribute. For example, if the sensitive attribute is a confounder, a control variable, or even a mediator. In each case, the relation would affect the fairness properties differently. Adding some discussion around this would be helpful.



**Summary Of The Paper:**

The paper investigates the challenges of in-processing fair algorithms with respect to accuracy and fairness in the setting of biased deployment data. A pre-processing step is proposed to mitigate this issue, along with an optimization approach.c

**Summary Of The Review:**

I tend to marginally accept this paper given the importance of the problem and the said approach.

---

> ### Author Response · Authors · 2022-11-16
> **Response to Reviewer gS56**
>
> We really appreciate your positive comments and thoughtful suggestions. We addressed them below and improved our manuscript.
>
> ----------------------
> **Comment 1: Causal perspective**
>
> We thank you for the great suggestion and fully agree that considering the different roles of the sensitive attributes would further enrich our discussion. We give a concrete example to show the correlation shifts when the sensitive group attribute itself is a confounder, one of the important roles of attributes in causality-based fairness.
> - We consider a car insurance example, where the goal is to predict future accident rates from the past driving record: suppose that X is the past driving record, Y is the future accident rate, and Z is the race (sensitive attribute).
> - Then, the structural causality graph between them would look like: Z -> (X, Y) and X -> Y, i.e., Z being a confounder. The arrows from Z to X and Y could be due to (a) different socialization and driving behavior formation process that partially depends on racial identity and (b) another unobserved mediator such as where they live in, etc. In this example where Z is a confounder, it is clear that Z and Y are correlated.
> - We expect this confounding effect to change over time. This will make the sensitive attribute Z either a stronger or weaker confounding factor, which impacts the correlation between Y and Z.
>
> We added this discussion and the causality graph in our revision (Sections 6 & C.3 and Figure 9, highlighted in blue).

---

### Official Review · Reviewer_H3mp · 2022-11-03

**Confidence:** 4
**Correctness:** 3
**Technical Novelty And Significance:** 3
**Empirical Novelty And Significance:** 3
**Recommendation:** 5

**Clarity, Quality, Novelty And Reproducibility:**

- Clarity and Quality: Contributions of the paper are clear and it is well-written.
- Novelty: The contributions of the paper are novel.
- Reproducibility: The authors have provided source code for experiments but I have not run them myself.

**Strength And Weaknesses:**

- Strengths:
1. The approach is a useful contribution to the field of fair ML since it is (a) well-motivated theoretically, (b) is a pre-processing step so advances in in-processing fair methods can still be utilized, and (c) the relaxed version of the pre-processing optimization problem is convex, ensuring "optimality" in approximating the true optimal.
2. The authors conduct extensive experiments to evaluate their approach with many correlation-shifted versions of datasets (training-set as Adult dataset (1996) with test-set as ACS income (2021), for example) which I believe can also be used as benchmarks for future work in this field. I also appreciate the inclusion of empirical results when the correlation shift range $[\alpha, \beta]$ is not known exactly (Section B.9), or misspecified (Section 5.3).
3. I believe the fact that the approach can work with multiple notions of fairness (such as Equalized Odds and Demographic Parity) is also a major strength.
4. The simulation results and toy examples discussed throughout the paper increase readability and understanding quite a bit.

- Weaknesses:
1. While the authors mention many related works under "Fairness-Specific Shifts", I am not quite sure why some of these approaches cannot be applied to the correlation-shift problem covered in the paper. Surely, subpopulation shifts (Maity et al, 2021) seem to be a more specific and constrained version of within-group correlation shifts, is there a way to link these together theoretically? Similarly, what are the exact differences between demographic shifts (Giguere et al, 2022) and correlation shifts? It might be beneficial if the authors can provide more details explaining the above, and possible even some theoretical/empirical evidence on the same.
2. Can the correlation shift study and pre-processing optimization approach studied in the paper have direct impact on the few existing works on poisoning attacks on fairness of classifiers [1-3]? As these attacks poison the training data, there is bound to be some correlation shift between the "original" test distribution (where the attacker would like the defender to fail) w.r.t the poisoned training distribution, and possible the pre-processing approach proposed can counteract the negative effects of these attacks. It would be interesting to see this discussed as well.


- References:
- [1] https://ojs.aaai.org/index.php/AAAI/article/view/17080
- [2] https://link.springer.com/chapter/10.1007/978-3-030-67658-2_10
- [3] https://link.springer.com/chapter/10.1007/978-3-031-00123-9_30

**Summary Of The Paper:**

The paper discusses the idea of enforcing fairness when training and testing/deployment distributions are not identical by making two contributions-- (i) the authors propose the notion of "correlation shifts" which analytically captures the aforementioned issue; (ii) the authors propose a pre-processing optimization approach that corrects for correlation shifts and existing in-processing approaches can then be used on the transformed data to ensure fairness. They conduct experiments on a number of baselines that show the efficacy of their approach and also propose dataset configurations with correlation shifts in testing.

**Summary Of The Review:**

I recommend this paper for acceptance based on the novel ideas proposed, the extensive experimental evaluation, and theoretical analysis. To strengthen the work further, there are a few papers related to this work for which the authors could provide some more comparative details as part of the discussion section.

__Edit__: After the discussion period with the AC and other reviewers, I have thought about some of the additional concerns raised. I am thus changing my score to borderline reject, but I do believe the work holds immense potential once it addresses these concerns. The main issue stems from the solution approach itself (I am summarizing from the AC/reviewer comments):
- The authors assume that a correlation constant between the training and deployment data exists. Thus, problem is not being solved. Instead, the "answer" is already included in the approach to achieve fairness in the deployment phase. The challenge in addressing the data bias change is how to estimate the correlation change-- even if it is chosen to be provided as a range instead of a precise number. It is non-trivial to utilize approaches such as those of (Huang et al., 2006; Zhang et al., 2013) for estimating the correlation factor range as these are not designed to handle the shift between the label and the sensitivity attribute.
- The experiments concerning unknown correlations, a large range of correlations, or the scenario when the knowledge of deployment data distribution is lacking, are conducted on a synthetic dataset, which might not coincide with real-world datasets.
- After some more thought, I am inclined to agree with Reviewer 9fri's concerns regarding the input distribution, and the scenario when the input distribution itself changes drastically.

---

> ### Author Response · Authors · 2022-11-16
> **Response to Reviewer H3mp**
>
> We really appreciate your positive comments and thoughtful suggestions. We addressed them below and improved our manuscript.
>
> -----------------------
> **Comment 1: Detailed comparison with other fairness-specific shifts**
>
> We thank you for this comment. The key difference between the correlation shift and the other fairness-specific shifts (subpopulation and demographic shifts) is that the other shifts are defined under specific assumptions on the data distribution, which are not required in our correlation shift definition. Here are the assumptions in the subpopulation and demographic shifts:
> - The subpopulation shift [1] assumes that the loss expectations w.r.t. input feature X and label Y of training and deployment distributions are the same (i.e., E_train[loss(h(X), Y) | Z=z] = E_test[loss(h(X), Y) | Z=z], where E indicates the expectation, and h($\cdot$) indicates the model output). Thus, as discussed in the related work section, a major example of subpopulation shifts is when a specific group has fewer positively-labeled examples during training time compared to deployment time, while the distributions of the X and Y attributes remain the same.
> - Similarly, the demographic shift [2] assumes that the joint probabilities of X and Y on the training and deployment distributions are identical (i.e., Pr_train(X=x, Y=y | Z=z) = Pr_test(X=x, Y=y | Z=z)). This assumption is similar to the assumption of the subpopulation shift, but the difference is whether the loss values of the model are explicitly considered or not.
>
> The above assumptions are used in the theoretical analyses of the previous works [1, 2]. In comparison, our theoretical analyses in Section 3 are not limited by any additional assumptions, and we show that the achievable performances of fair training are determined by the (y, z)-correlation.
>
> We added the above additional discussion in our revision (Section C.2, highlighted in blue).
>
> [1] Maity et al., “Does enforcing fairness mitigate biases caused by subpopulation shift?”, NeurIPS’21
>
> [2] Giguere et al., “Fairness guarantees under demographic shift”, ICLR’22
>
> -----------------------
> **Comment 2: Algorithm behavior on poisoning attacks on fairness**
>
> As per your great comment, we evaluated the performance of our method on poisoned data and observe that our method can potentially be extended to the poisoning attack scenario.
>
> We performed a new experiment using one of the poisoning attack methods for fair training [1], where 10% of the training labels of a specific group are flipped so as to maximize the accuracy degradation. The table below shows the accuracy and fairness performances of the in-processing-only baseline FairBatch (FB) and our framework w.r.t. demographic parity (DP) on the poisoned data; we also report FB on the clean data, which can be considered as the upper-bound performance. Here, when achieving similar fairness, our framework improves the accuracy of FB on the poisoned data. We suspect that, while our pre-processing reduces the correlation shift between training and deployment data, the effect of the poisoning attack is also mitigated.
>
> We note that these results are preliminary and only show that our method has some potential to support the poisoning attack scenario, and we believe there is plenty of room for improvement. In particular, we believe our method can be further extended with other fair and robust training methods like [1].
>
> We added these results and discussion in our revision (Section 6 and Section B.14, highlighted in blue).
>
> [1] Roh et al., FR-Train: A Mutual Information-Based Approach to Fair and Robust Training, ICML’20
>
> |     | Acc. | DP disp.|
> |:--------------:|:-----------------------:|:------------------------:|
> |FairBatch (in-processing-only) on the poisoned data|0.699|0.026|
> |Ours + FairBatch on the poisoned data| 0.721 |0.024|
> | | |
> |FairBatch on the clean data *(upper bound)* |0.773|0.023|
> | | |

---

> > ### Comment · Reviewer_H3mp · 2022-11-17
> > **Response to Authors**
> >
> > Thank you for the additional experiments on poisoning attacks as well as the detailed clarifications regarding the other fairness specific distribution shifts. I think the current score remains fair, given that this is a solid paper with potential for significant impact in a subfield of the fairness community.

---

> > > ### Author Response · Authors · 2022-11-17
> > > **To Reviewer H3mp**
> > >
> > > We really appreciate your response and the positive appraisal.
> > >
> > > We are also very happy that our responses have addressed your comments. Thank you again for your valuable suggestions, including 1) the detailed comparison with other distribution shifts and 2) the possible extension to the poisoning attack scenario, which indeed helped to improve the manuscript.

---

> ### Author Response · Authors · 2023-01-12
> **Additional Response to Reviewer H3mp (Part 1)**
>
> We appreciate your additional comments, which helped us to improve our manuscript. We do believe the concerns are addressable: 1) we can reasonably estimate the shifted correlation to run our algorithm, and 2) our framework can handle real-world x distribution changes. We verify our claims via additional experiments on real-world benchmark datasets.
>
> We note that the concern about estimating the shifted correlation seems to have been brought up after the discussion period, which we were not aware of, and we believe we should have a fair chance to address it.
>
> **Estimating the shifted correlation:**
>
> We first explain how we can detect the correlation shifts to run our algorithm. To estimate the shifted correlation, we consider the scenario where accessing some of the new data is possible, which is a common and practical setting for detecting drifts [1] in a data stream. In this setup, one needs to look at the new data to even realize that there is a drift. Here, using the small amount of new data, one can estimate the new correlation by calculating the empirical probability with the given samples. Alternatively, one can use distribution estimation techniques [2, 3] for computing the correlation and generating a range using its confidence interval, as we discussed in our paper; however, we will show below that we can get reasonable results even without such estimation techniques by only accessing a very small amount of new data.
>
> **Handling x distribution changes:**
>
> We would also like to clarify that 1) the assumption on the x distribution is reasonable, and 2) our algorithm still performs well even when the x distribution changes.
>
> - In distribution shift studies [2, 4, 5, 6], it is common to assume that the training and deployment data distributions differ only in specific aspects (e.g., only label distribution is changed). Without such an assumption, the training and deployment data distributions can be arbitrarily far apart, and there is no way to infer the deployment distribution via the training samples [2]. In our framework, we focus on the distribution changes of y and z and do not explicitly model the change of the input feature x distribution.
> - Our framework thus works best when the x distribution does not change; however, ours does not strictly require this condition, unlike other previous works on fairness under different types of shifts [5, 6]. Indeed, we empirically show that our framework performs well in a real-world scenario where the x distribution does shift (see below).

---

> > ### Author Response · Authors · 2023-01-12
> > **Additional Response to Reviewer H3mp (Part 2)**
> >
> > **Additional experiments to support our clarifications:**
> >
> > To verify the above discussions in a real-world scenario, we evaluate our framework when 1) the shifted correlation is estimated from only a very small amount of deployment data and 2) when the x distribution changes. We use the two real-world income datasets collected in the 1990s (i.e., AdultCensus [7]) and 2010s (i.e., ACSIncome [8]) for training and testing, respectively. Since the two datasets have a large time gap, they naturally have different (y, z)-correlations as well as input feature x distributions (see Figure 5 of our paper for an analysis). Following the scenario discussed above, we assume access to only 1% of the 2010s data to measure the shifted correlation constant c. Note that accessing a small amount of the new data is a common setting in detecting drifts [1]. With the 1% data, we compute the difference between empirical probability P(y|z) values, as defined in Sec. 4.1. As a result, the estimated c is 0.159, where the true c computed on the entire new data is 0.147. When running our pre-processing with the estimated c, our framework still outperforms the in-processing baselines, as shown in the table below. As a result, there are **two takeaways**: 1) one can reasonably estimate the correlation shift via a small amount (say 1%) of deployment data, and 2) our method performs well with the estimated correlation, even with x distribution shifts.
> >
> > In our revised paper (although we are not allowed to upload to OpenReview now), **we have reflected all the above clarifications and experiments** by adding new paragraphs, sections, and remarks. We will release the new version of the paper when the update is available.
> >
> > -----------------------
> > [Table. Model performances on the ACSIncome test data collected in the 2010s. The models are trained on the AdultCensus data collected in the 1990s. **Note**: We also used these datasets in our existing experiment (Sec B.7), but the setting is now different as we use the estimated correlation value rather than the true correlation value.]
> >
> > |   Method     |     Accuracy    | Unfairness (DP) |
> > |---------------|-----------------|-------------------|
> > |   FC             | 0.646 ± 0.013 |  0.104 ± 0.015  |
> > |   **Ours**+FC   | 0.661 ± 0.013 |  0.086 ± 0.014  |
> > | | | |
> > |   AD             | 0.637 ± 0.031 |  0.137 ± 0.037  |
> > |   **Ours**+AD   | 0.634 ± 0.021 |  0.063 ± 0.037  |
> > | | | |
> > |   FB             | 0.643 ± 0.007 |  0.128 ± 0.021  |
> > |   **Ours**+FB   | 0.673 ± 0.005 |  0.091 ± 0.008  |
> > | | | |
> > |   FB on test dist. (*upper bound*)   | 0.704 ± 0.007 |  0.044 ± 0.035  |
> > | | | |
> >
> >
> > -----------------------
> > [1] Lu et al., “Learning under concept drift: A review”, IEEE TKDE 2018.
> >
> > [2] Huang et al., “Correcting sample selection bias by unlabeled data”, NIPS 2006.
> >
> > [3] Zhang et al., “Domain adaptation under target and conditional shift”, ICML 2013.
> >
> > [4] Chen et al., “Fairness Transferability Subject to Bounded Distribution Shift”, NeurIPS 2022.
> >
> > [5] Maity et al., “Does enforcing fairness mitigate biases caused by subpopulation shift?”, NeurIPS 2021.
> >
> > [6] Giguere et al., “Fairness guarantees under demographic shift”, ICLR 2022.
> >
> > [7] Kohavi, “Scaling up the accuracy of naive-bayes classifiers: A decision-tree hybrid”, SIGKDD 1996.
> >
> > [8] Ding et al., “Retiring adult: New datasets for fair machine learning”, NeurIPS 2021.

---

### Author Response · Authors · 2023-02-07
**To general readers: Updates after the ICLR decision**

We have further updated our manuscript based on the meta-review from ICLR. The revised version of the paper can be found in **https://arxiv.org/abs/2302.02323**.

The key changes are as follows:

1) We clarified **how to estimate the shifted correlation value** in our framework.
   - Specifically, we added an explicit step for estimating the shifted correlation in the scenario of when some (say $m$ samples) of the new data is available, which is a common and practical setting for detecting data drifts [1]. In this scenario, we can estimate the new correlation constant $c$ by calculating the empirical probability $\hat{\Pr}(y|z)$ using the given $m$ samples via a maximum likelihood estimator.
   - We can also compute a confidence interval of the estimated $c$ using standard concentration inequalities (e.g., Hoeffding’s inequality). We gave a *formal estimation guarantee* and *proof* using Hoeffding’s inequality in the Appendix of our revision.
   - In our experiments, we also showed that 1) the correlation can be reasonably estimated via small amounts of new data and 2) our framework works well even when the estimation is slightly off (see the next section for more details).
2) We performed **more experiments on real-world datasets to show 1) the robustness against misspecifying correlation and 2) performances on x distribution shifts of our algorithm.**
   - We showed that our framework is still beneficial when the correlation estimation error is reasonable (say ± 10%). In this experiment, we use both synthetic and real-world COMPAS datasets [2].
   - We also empirically showed that our framework performs well in a real-world scenario where even the x distribution changes in addition to the (y, z)-correlation shifts. In this experiment, we use two real-world income datasets collected in the 1990s (i.e., AdultCensus [3]) and 2010s (i.e., ACSIncome [4]) for training and testing, respectively.
3) We added **more discussions on the recent related works and evaluated our framework in several new scenarios** (e.g., showing some potential to support the poisoning attack scenario), which have been requested during the ICLR rebuttal period.

We again appreciate the reviewers and area chair for their comments and believe the remaining issues have been addressed in our revision.

--------------------
[1] Lu et al., “Learning under concept drift: A review”, IEEE TKDE 2018.

[2] Angwin et al., “Machine bias: There’s software used across the country to predict future criminals. And its biased against blacks”, ProPublica, 2016.

[3] Kohavi, “Scaling up the accuracy of naive-bayes classifiers: A decision-tree hybrid”, SIGKDD 1996.

[4] Ding et al., “Retiring adult: New datasets for fair machine learning”, NeurIPS 2021.

---

### Decision · Program_Chairs · 2023-01-20

**Decision:**

Reject

**Justification For Why Not Higher Score:**

This paper is not self-standing. The research problem is to tackle the data bias change, where the authors directly assume a correlation constant between the training and deployment data. From a philosophy perspective, the authors do not solve the research problem. Instead, they incorporate the answer as input to achieve fairness in the deployment phase.

**Justification For Why Not Lower Score:**

N/A

**Metareview: Summary, Strengths And Weaknesses:**

Based on the collected information from all reviewers and my personal judgment, I can make the recommendation on this paper, **rejection**. In sum, this paper is in good shape. Here are the comments that I summarized, which include my opinion and evidence.

**Research Problem**

The authors argue that it is a strong assumption that the training and deployment data distributions are the same. In light of this, the authors aim to address the data bias changes.

**Motivation**

To further analyze the data bias (not the data bias change), the authors introduce the notion of correlation shift between the label and the sensitivity attribute. From the results of simulations in Section 3, the authors demonstrate low correlation enables classifiers to attain better accuracy-fairness tradeoffs. Unfortunately, there is a gap between the data bias and data bias change.

**Philosophy**

To tackle the data bias change, the authors directly assume a correlation constant between the training and deployment data. From a philosophy perspective, the authors do not solve the research problem. Instead, they incorporate the answer as input to achieve fairness in the deployment phase. The challenge in addressing the data bias change is how to estimate the correlation change. This is my major concern. I cannot follow the impractical setting of this paper, even though the authors relax the correlation constant in a range.

Beyond that, Reviewer 9fri points out that the authors do not consider the shift of the input feature.

**Technique**

Technically, the proposed method can be regarded as a pre-processing technique, then followed by an in-processing algorithm for unfairness mitigation.

**Experiments**

(1) Hyperparameter. The authors mention some methods (Huang et al., 2006; Zhang et al., 2013) that can estimate the correlation constant range. However, they are not designed to handle the shift between the label and the sensitivity attribute.

(2) All the experiments on handling unknown correlations or a large range of correlations are conducted on a synthetic dataset.

(3) The authors claim that their method can still work when the knowledge of deployment data distribution is lacking and demonstrate some results on a synthetic dataset in the paper. Actually, this is not the full picture. Synthetic datasets are usually used to verify the applied conditions of the proposed method. It is a plus to demonstrate some failure cases, which are quite helpful to understand the applied conditions.

**Minor**

Personally, the title “The World is Changing” is not informative.

No objection was raised from the reviewers on the rejection recommendation.